# Foundation Models for Industrial Scheduling Leveraging the Techniques from LLMs

## Abstract

The advent of large language models (LLMs) has significantly boosted productivity across various sectors. However, their application in the industrial domain remains underexplored and often yields suboptimal results, primarily due to stringent requirements for technological maturity, safety, and standardization. To address this gap, we leverage key techniques instrumental to the success of LLMs—such as the decoder-only architecture and scaling laws—rather than using LLMs directly, to develop a foundational model for industrial scheduling. In contrast to prior methods that focus on specific types of scheduling problems, our model is designed as a general-purpose framework capable of handling mixed operation types, objectives, and constraints reflective of real-world industrial environments. Through extensive experiments, our foundation models have demonstrated clear superiority over conventional scheduling methods and algorithms using LLMs directly. Notably, the foundation models for scheduling have exhibited scaling law, generalization ability, and adaptability analogous to those observed in LLMs. These results indicate that the principles underpinning LLMs extend beyond natural language processing, showing strong potential for broader industrial and manufacturing applications. Code at `https://anonymous.4open.science/r/Foundation-Models-for-Industrial-Scheduling-7BD4`

## 1 Introduction

The advent of Large Language Models (LLMs), such as OpenAI's GPT series Achiam et al. (2023), Meta's Llama Touvron et al. (2023), and Deepseek Liu et al. (2024), has marked a transformative milestone in the field of Natural Language Processing (NLP). By undergoing training in vast and diverse text datasets, LLMs have effectively captured the underlying principles of linguistic rules and semantic logic. LLMs have exerted a revolutionary impact on the field of NLP, even successfully realizing a portion of the aspirations previously held for artificial general intelligence (AGI). Consequently, LLMs have substantially improved productivity across a range of tasks, including code programming and content creation.

Owing to their advanced capabilities, LLMs have increasingly been investigated for potential applications across diverse domains, in particular in industry and manufacture. However, application of LLMs in industry is hindered by the high demands for technological maturity, safety, and standardization in real-world industrial environment Rane (2023); Ren et al. (2025). Specifically, three major challenges exist: 1) Reliability requirements: LLMs based on probabilistic models lack uniform strict requirements for output precision and reliability Rawte et al. (2023); 2) Limited domain-specific data: Industrial datasets are highly limited, particularly for specific industrial sectors; 3) Specialized vs General: As general-purpose models, LLMs demonstrate strong capabilities in handling broad knowledge-based queries, but the industrial challenge is a series of specialized issues.

In response, we propose to adapt techniques emerged from the rapid evolution of LLMs to solve practical industrial problems compared to use LLMs directly, such as the decoder-only architecture Radford et al. (2018), the scaling law Kaplan et al. (2020), reinforcement learning Christiano et al. (2017) and so on.

Existing scheduling approaches have predominantly focused on addressing specific categories of the job shop scheduling problem (JSSP), often overlooking the fact that real-world industrial environments typically involve diverse characteristics, constraints, and objectives Xiong et al. (2022b). To bridge this gap, we aim to develop a general foundational model for industrial scheduling. Specifically, it will be trained on the flexible job shop scheduling Problem (FJSP) and subsequently adapted to various real-world industrial scheduling tasks through fine-tuning.

In conclusion, we leverage techniques from LLMs, tailored to the specific requirements of industrial scheduling, to construct foundational models for industrial scheduling. Experimental results demonstrate that our foundation models yield superior performance in terms of solution quality and efficiency, and inherits key characteristics of LLMs, including notable scalability, generalization capability, and adaptability. Furthermore, to our knowledge, we present the first empirical evidence showing that the scaling law also holds for industrial problems. Our results indicate that the technologies and methodologies developed from the advancement of LLMs can hold significant potential across other critical domains as well, such as industrial scheduling sectors. Moreover, instead of stiffly applying LLMs to specific problems, it is more effective to integrate LLM-inspired design principles with domain-specific characteristics, ensuring a fusion of techniques and requirements.

## 2 RELATED WORK

The integration of AI into industrial applications has been a transformative force across various sectors, revolutionizing traditional manufacturing processes and enabling unprecedented levels of efficiency, adaptability, and automation Wan et al. (2021); Peres et al. (2020). Currently, traditional discriminative AI have been successfully implemented in industry to improve productivity due to their technological maturity and safety. For example, computer vision is employed to automate the detection of product defects Ren et al. (2022), and machine learning is used to forecast potential equipment failures during production processes Leukel et al. (2021). Emerging generative AI technologies, particularly LLMs, still face significant challenges in direct industrial deployment.

However, in the context of the growing significance of industry, where nations are vying to introduce new industrial plans Lasi et al. (2014); Xu et al. (2021); Wübbeke et al. (2016); Holdren et al. (2012), the application of state-of-the-art AI technologies in industry has become increasingly important. Notably, the application of LLMs has emerged as a key research focus in academia, with numerous studies exploring their potential for industrial production enhancement Wang et al. (2024); Javaid et al. (2023); Rane (2023); Ren et al. (2025). In engineering design tasks, multi-modal LLMs have demonstrated promising capabilities Picard et al. (2023); Wang et al. (2024). Additionally, Extensive studies have explored LLM applications in specialized industrial fields, such as power systems and chemical engineering Majumder et al. (2024); Hamann et al. (2024); Decardi-Nelson et al. (2024), yet the proposed domain-specific integration frameworks largely remain conceptual. Regarding industrial scheduling problems, numerous studies have employed AI Zhang et al. (2020); Peres et al. (2020); Xiong et al. (2022b). However, a predominant limitation in existing approaches is their treatment of scheduling as an example of combinatorial optimization problems, often neglecting critical industrial characteristics. Recent attempts to apply LLMs to industrial scheduling Abgaryan et al. (2024; 2025) have been constrained by their reliance on supervised learning paradigms, resulting in performance that largely reflects the limitations of the training data rather than real-world industrial application.

In conclusion, while leveraging LLMs to empower industries has become a consensus among researchers and practitioners, industrial demands for technological maturity, safety, and standardization render this vision highly challenging. Further research is needed to facilitate the practical deployment of LLMs in industrial scenarios.

## 3 PRELIMINARIES

**Flexible Job-shop Scheduling problem**. FJSP demonstrates strong applicability to complex scheduling scenarios under real-world operating conditions. While most existing studies focus solely on machine flexibility, this paper considers a more comprehensive FJSP model that incorporates two types of flexibility: alternative routings and alternative machines, as illustrated in Fig. 1.

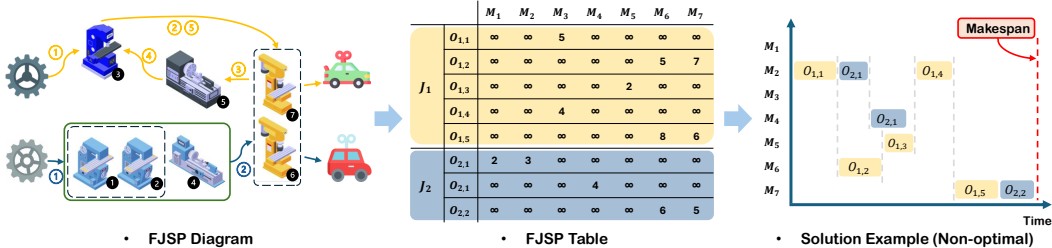

Figure 1: Schematic of an FJSP example: 1) The diagram of FJSP; 2) The corresponding table data form of FJSP; 3) A solution example (non-optimal).

In detail, a standard FJSP can be stated as follows. A set of $J = \{J_1, ..., J_i, ..., J_n\}$ of $n$ jobs. A set of $M = \{M_1, ..., M_j, ..., M_m\}$ of $m$ machines. Each job $J_i$ consists of a specific sequence of $l_i$ consecutive operations $O_i = \{O_{i,1}, ..., O_{i,k}., ..., O_{i,l_i}\}$ with precedence constraints. Note, different operations could share same sequence index $k$, which indicates that the order between them can be flexibly arranged. Furthermore, let $T_O^{i,j,k}$ represent the processing time of operations $O_{i,k}$ on the machine $M_j$. An operation $O_{i,k}$ could only be executed on a subset of eligible machines $M_{i,k} \subseteq M$. Thus, for convenience, $T_O^{i,j,k}$ would be set to $+\infty$ if the operation $O_{i,k}$ cannot be executed on machine $M_j$. Each operation cannot be interrupted once started, and one machine can only perform one operation at a time. Besides, all machines and jobs are available in the beginning of the time horizon. In this work, the basic objective of FJSP is to minimize the maximum completion time of all jobs, that is, minimizing the total makespan $C_{\max} = \max_{1 \leq i \leq n} C_i$, as shown in Fig.1.

FJSP serves as a generalized framework for industrial scheduling problems, which can be transformed into various scheduling problems by imposing additional constraints to model real-world industrial scheduling problems. For example, when each operation is restricted to one single machine, the problem reduces to the classical Job-shop Scheduling Problem Xiong et al. (2022a). If all jobs share identical processing sequences, the problem exhibits characteristics of the Non-permutation Flow-Shop Scheduling Problem Rossit et al. (2018). When operations have not strict order limitation, the problem demonstrates features of the Open-Shop Scheduling Problem Anand et al. (2015). Therefore, developing a foundational model for FJSP could provide a unified scheduling solution adaptable to various industrial domains.

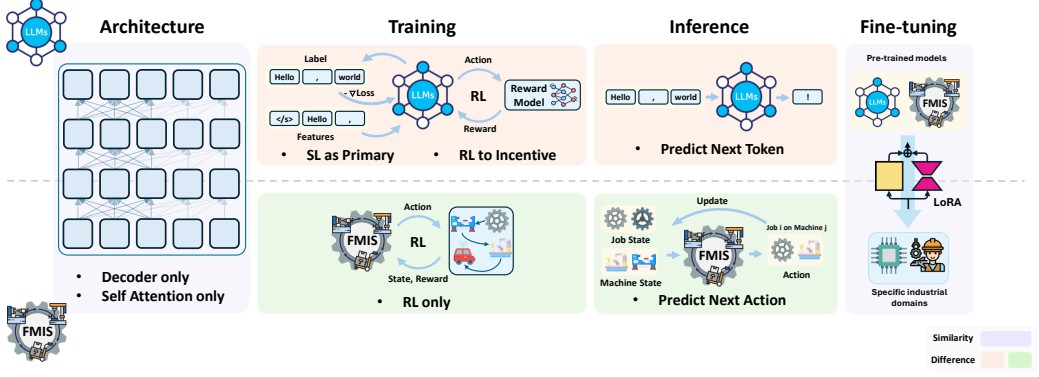

Figure 2: Comparison of foundation models for industrial scheduling (FMIS) and LLMs, highlighting key similarities and differences.

## 4 PROPOSED METHOD

In this section, we describe the framework and implementation details of foundation models for industrial scheduling. To highlight the key characteristics of our approach, we systematically compare it with LLMs in three aspects: architecture, training, and inference, as illustrated in Fig. 2.

### 4.1 ARCHITECTURE

From the inception of LLMs, the decoder-only architecture has played a pivotal role, serving as the cornerstone of the remarkable advancement of LLMs Radford et al. (2018). The decoder-only architecture eliminates the encoder component and avoids computational redundancy from bidirectional attention mechanisms, thereby enabling more efficient utilization of computing resources. This design makes distributed training feasible for extremely large-scale models. The decoder-only structure serves as the basement for LLMs to exhibit the scaling law —- the emergent capabilities in reasoning and generalization that manifest only beyond certain parameter thresholds Kaplan et al. (2020); Wei et al. (2022).

Therefore, our foundation models for industrial scheduling adopt this decoder-only architecture, as illustrated in Fig. 3, which comprise multiple Transformer decoder layers, each containing masked multi-head self-attention networks and feed forward neural networks (FFNs), enhanced by residual connections and layer normalization for stable gradient flow Vaswani et al. (2017).

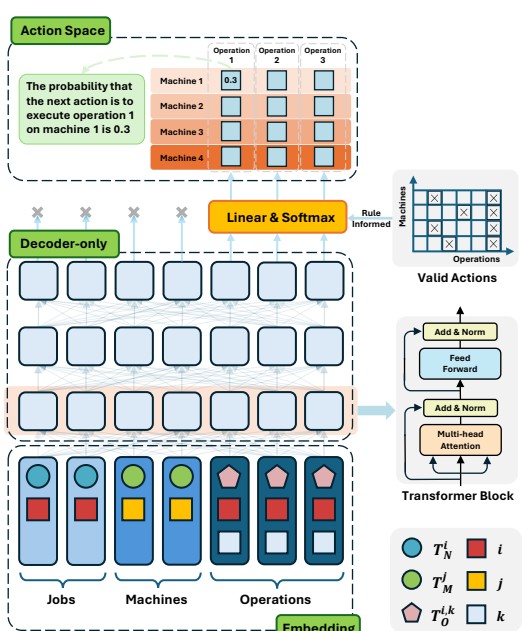

Considering the fundamental differences between industrial scheduling and NLP, our models distinguish with LLMs in several aspects. While LLMs employ causal masking to restrict each position's access to only current and historical information —aligning with the sequential nature of text generation Vaswani et al. (2017), our models implement modified causal masking that ensures only state information relevant to solving the current scheduling problem could be accessed. Specifically, it filters out selected operations. Furthermore, while LLMs rely on positional encoding to explicitly capture sequential order (addressing the position insensitivity of pure attention mechanisms) Vaswani et al. (2017), our foundation models for industrial scheduling utilize inherent indexes of jobs, machines, and operations as encoding inputs, as shown in Fig. 3. These designs better accommodate the structural characteristics of industrial scheduling problems.

Figure 3: Architecture of foundation models for industrial scheduling.

### 4.2 TRAINING

Training of LLMs is predominantly conducted during the pre-training phase, involving training on vast amounts of text data Vaswani et al. (2017). On this basis, reinforcement learning only serves to reinforce or incentivize based on human preferces in LLMs Christiano et al. (2017).

However, as previously discussed, industrial datasets are inherently more limited compared to other domains. Therefore, we primarily employ RL to train our foundation models for industrial scheduling. Our foundation models would learn by interacting with the industrial scheduling environment, receiving rewards or punishments based on its actions.

We adopt Proximal Policy Optimization (PPO), which is a typical RL algorithm employed in LLMs. The PPO framework consists of a actor network (Fig. 3), which outputs the scheduling policy, and a critic network that estimates expected returns. The detailed principles of PPO are provided in Appendix A.1. In our implementation, the critic network shares the same architecture as the actor network, except that the final softmax function is replaced with a sum function. The reward function is defined as the amount of change in makespan after each action, namely $r_t = -\Delta C_t$, where $C_{\max} = \sum_{t \in \tau} \Delta C_t$.

### 4.3 INFERENCE

The inference process of LLMs is fundamentally based on auto-regressive generation. LLMs generate text token-by-token, where each step predicts the next token $x_t$ conditioned on the preceding sequence $x_{1:t-1}$, following the distribution $P(x_t|x_{1:t-1})$. Sampling strategies (e.g., greedy search, beam search, or nucleus sampling) determine token selection Shi et al. (2024). The inference mechanism of LLMs is fundamentally designed under the assumption that each token depends only on its preceding context, a principle that aligns with the sequential nature of textual data, thereby leading an advantage over BERT Devlin et al. (2019).

Drawing inspiration from LLMs, we design the inference framework of our foundation models based on a same causal dependency. The inputs for foundation models for industrial scheduling problems only include the following components: 1) Job State ($T_N^i$): Indicates the completion time of the last scheduled operation of the job $J_i$; 2) Machine State ($T_M^j$): Indicates the completion time of the last scheduled operation on the machine $M_j$; 3) Operation: Describes pending operations awaiting scheduling, where $T_O^{i,k} = \{T_O^{i,1,k}, \ldots, T_O^{i,m,k}\}$ defines the processing time of operation $O^{i,k}$ on each machine.

Given these inputs, our foundation models would predict the next action. Specifically, the model outputs a probability matrix $P_{(n,l) \times m}$, where $p_{(i,k),j}$ represents the probability of $O_{i,k}$ being executed on $M_j$. as shown in Fig. 3.

**Rule Informed**: To address the high unreliability of probabilistic predictions, our foundation models employ scheduling rules to directly constrain the output space, thereby ensuring the feasibility of scheduling solutions to meet the stringent reliability requirements of industrial applications. Specifically, we derive a feasibility matrix $F(S)$ from the scheduling rules (e.g., sequential constraints on operations, machine failures, job cancellations, etc.) to indicate the feasibility of each corresponding action. This matrix is then incorporated into the output layer during the Linear&Softmax process according to Eq. 1.

$$P_{(n,l) \times m} = \text{Softmax}(\text{Linear}(x) - \overline{F(S)} \cdot \infty) \tag{1}$$

The beam search sampling strategy is utilized for selecting the action based on $P_{(n,l) \times m}$ Shi et al. (2024).

### 4.4 FINE-TUNING

Similar to how LLMs are fine-tuned for domain expertise, we apply LoRA Hu et al. (2022) to our foundational model. This fine-tuning enables superior performance when applied to various shop scheduling scenarios with differing constraints, objectives, and practical conditions.

To address the new conditions introduced by variations in these industrial scheduling problems, we adopt a concept similar to ControlNet Zhang et al. (2023). This involves creating a duplicate module alongside the original one. The duplicate is connected to the original via a linear layer whose initial weights are set to zero, thereby introducing new conditions while preserving the stability of the pre-trained model parameters, as shown in Fig 4. To incorporate new constraints, we add a weighted violation penalty to the reward function and progressively increase the weight during fine-tuning until constraints are satisfied. For new objectives, we decompose the problem using the Chebyshev method, formulated as $g(x|w) = \max_{i=1,\ldots,m} w_i|f_i(x) - z_i^*|$. By treating the weight vector $w$ as a model input, we can obtain the complete Pareto front by varying $w$ during deployment.

In addition to the aforementioned techniques, technologies such as mixed-precision computing Micikevicius et al. (2017), FlashAttention Dao et al. (2022), and parallelism of data and model Rasley et al. (2020), which emerged during the development of LLMs, have also been employed in foundation models for industrial scheduling, significantly enhancing both training and inference efficiency.

## 5 EXPERIMENT

### 5.1 EXPERIMENTAL SETTINGS

**Datasets**: An FJSP instance with $n$ jobs and $m$ machines is denoted as "$n \times m$" in short. As in most related work, we generate synthetic FJSP instances with a wider range for the random processing time for pre-training and testing. For a fair and comprehensive comparison, we consider seven different scales of FJSP instances from $10 \times 5$ to $50 \times 10$. Without additional specification, our models are trained on instances $10 \times 10$ only. The training data are randomly regenerated at each episode to avoid overfitting.

**Configurations**: In order to observe the scaling laws in the industrial scheduling base model, we trained a series of models with different scales. Referring to previous studies on scaling effects in LLMs, the number of parameters ranges from $10^5$ (XXXS) to $10^9$ (XXL). The specific model hyperparameters are provided in Appendix A.2. All experiments were performed on a computing device equivalent to 8 NVIDIA H800 GPUs.

**Comparative Algorithms**: To comprehensively evaluate the performance of our model, this paper will include the following comparative algorithms: 1) Priority dispatching rules (PDRs) Haupt (1989); 2) Learning-based algorithms: DRLZhang et al. (2020) and MARL Lei et al. (2022); 3) LLMs-based algorithms: OPRO Yang et al. (2024) and GEN Abgaryan et al. (2024); 4) Meta-

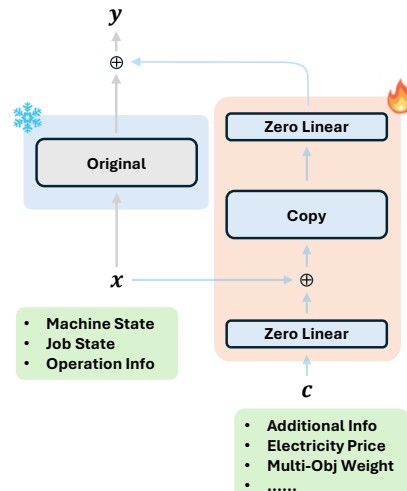

Figure 4: Methods for adding additional inputs in ControlNet.

heuristic algorithms: 2SGA Defersha & Rooyani (2020), HQPSO-VNS Xu et al. (2024), NSGA-II Deb et al. (2002), and MOEA/D Zhang & Li (2007); Xiao et al. (2024). These algorithms will be compared with our model based on their applicability to various problem domains. The details of following comparative algorithms in this paper are provided in Appendix A.3.

The relative gap is used to evaluate the quality of the solution. For each solution with makespan $C_{\max}$, its relative gap to the makespan $C_{\max}^{\text{BS}}$ of the best known solution (not necessarily optimal) is calculated as $(C_{\max}/C_{\max}^{\text{BS}} - 1) \times 100\%$. When multiple objectives are involved, hypervolume (HV) is used to evaluate.

### 5.2 RESULTS AND DISCUSSION

#### 5.2.1 VALIDATION OF THE SCALING LAW

The scaling law serves as a fundamental guideline in LLMs development, informing both theoretical frameworks and engineering practices. To investigate whether a similar scaling law governs our models, we evaluated the performance of different model scales and their corresponding convergence curves. As shown in Fig. 5, our models exhibit a partial alignment with scaling laws, showing performance trends that improve with training compute, model size, and environmental interaction (analogous to dataset size in LLMs). However, unlike LLMs, the performance does not improve indefinitely with scale. Once the model exceeds 500M parameters, there is not significant performance difference.

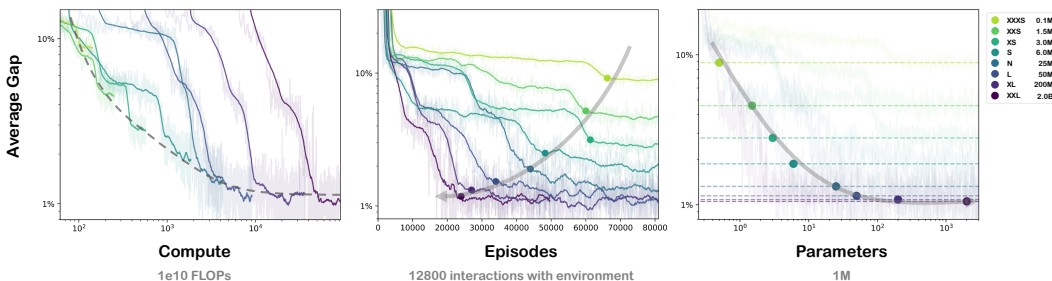

Figure 5: Impact of model size, environmental interactions, and training compute on the performance of foundation model for industrial scheduling.

Besides the final performance, the scale of the model also significantly impacts its learning speed. As illustrated in Fig. 5, the number of episodes required for model convergence decreases with increasing model scale, indicating that larger models require fewer environmental interactions. We posit that this phenomenon occurs because larger models can memorize more scenarios, thereby achieving more efficient and stable learning.

### 5.2.2 PERFORMANCE ON BENCHMARKS

We performed experiments on randomly generated FJSP instances. As summarized in Fig. 6 and Tab. D-2, our foundation model (N) demonstrates superior performance, stability, and generalization capability compared to the best PDR (FIFO-EFT) on the problem and learning-based algorithms (DRL, MARL). The huge advantage of our foundation model over learning-based algorithms suggests that, rather than employing specialized model architectures, a simpler decoder-only structure with scaled-up parameters would prove more effective — a finding consistent with the developmental trajectory of LLMs. However, algorithms that directly LLMs (OPRO, GEN) often suffer from suboptimal performance, low efficiency, and unstable results. In particular, OPRO's approach of employing LLMs as optimizers results in a significant waste of tokens.

In terms of computational efficiency, although our approach involves much higher computational complexity than PDRs and leaning-based algorithms, this is mitigated by the parallel processing capabilities and memory optimization benefiting from the rapid advancements driven by AI, especially LLMs. Therefore, as the results show in Fig. 6, our model is far more efficient than the rest of the algorithms that use large models, and is on the same level as the learning-based algorithms.

Compared to OR-tools, the proposed model rapidly achieves comparable or superior results to OR-Tools on FJSP benchmark instances (HurinkVdata 19, 29, 34) Hurink et al. (1994), as shown in Fig.

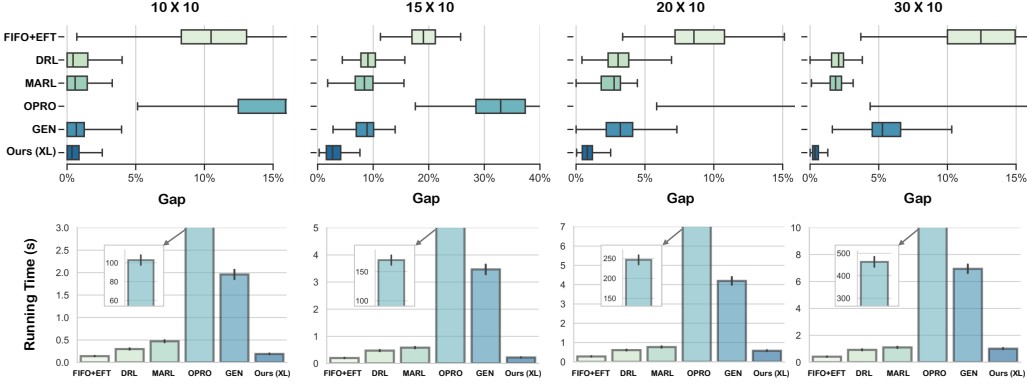

Figure 6: Statistical results of our foundation model for industrial scheduling and comparative algorithms on randomly generated instances, including gap and running time.

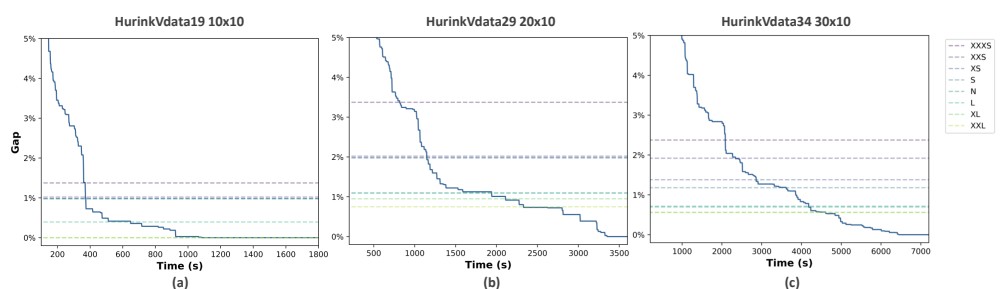

Figure 7: Comparison of OR-Tools optimization convergence and our foundation model for industrial scheduling performance on FJSP Hurink's instances with different scale.

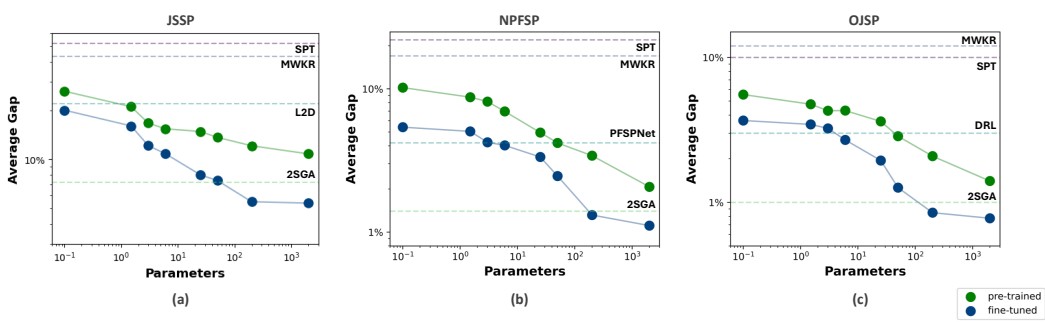

Figure 8: Performance of the foundation model for industrial scheduling across different scheduling problems: a) JSSP; b) NPFSP; and c) OJSP.

7. Our model reaches the optimization quality within seconds which QR-tools require thousands of seconds, highlighting its efficiency advantage that grows with problem size. The parameters employed by OR-tools are provided in the Appendix A.4.

The complete experimental results, such as comparative experiments on the standard benchmarks (hurink's instances ) and other comparative algorithms, are provided in Appendix A.4.

### 5.2.3 PERFORMANCE ON VARIANTS

Our foundation models are further evaluated on additional scheduling problem types (JSSP, NPFSP, and OJSP) with Taillard's instances Taillard (1993). Besides, our foundation models would be tested on industrial scheduling variants with distinct constraints and objectives. Here are the chosen

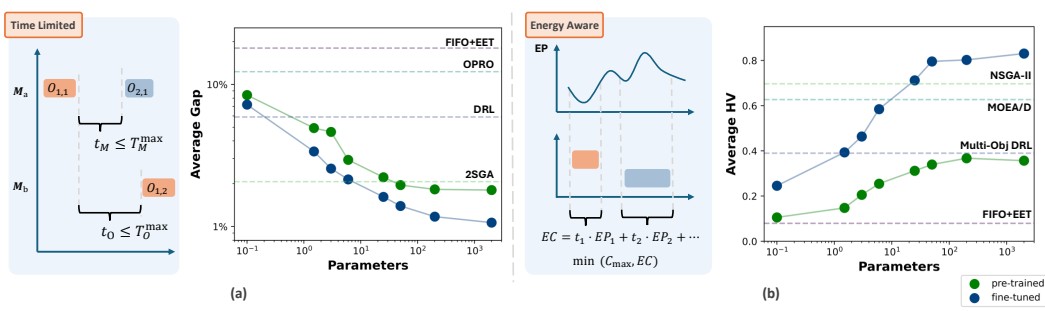

Figure 9: The schematic diagrams of industrial scheduling variants and comparative results. a) Time limited FJSP; b) Energy-aware multi-objective FJSP.

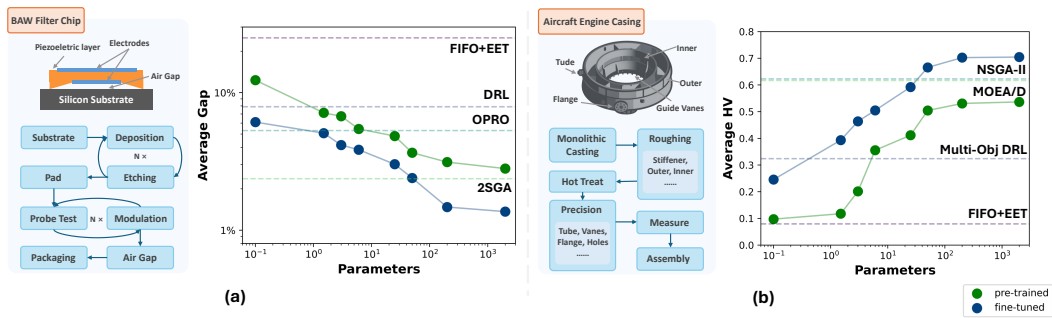

Figure 10: The schematic diagrams of real-world industrial scheduling examples and comparative results. a) BAW filter chip production; b) aircraft engine casing production.

variants: 1) Time limited: Maximum operation waiting time and maximum machine idle time, reflect practical requirements like maintaining workpiece integrity or machine operating temperatures; 2) Energy-aware: minimize both the makespan and the total electricity cost May et al. (2015). We employ LoRA Hu et al. (2022) to fine-tune our model, with a rank of 8 and 4,000 training episodes, while keeping all other parameters aligned with pre-training settings.

As Fig. 8 and 9, our foundation models demonstrate consistent and significant advantages over the compared algorithms, both before and after fine-tuning. Even when metaheuristic algorithms (2SGA, NSGA-II, MOEA/D) are granted a substantially longer runtime, our model ($\geq$ XL size) achieves superior results. These results highlight that our models possess the key attributes expected of foundation models in scheduling tasks.

### 5.2.4 PERFORMANCE ON REAL-WORLD INDUSTRIAL CASES

We select two real-world industrial scheduling cases from aerospace manufacturing, as shown in Fig. 10, to further evaluate our models. The details of these cases are provided in Appendix A.2. As shown in Fig. 10, our models outperform all comparative methods overall, exhibiting strong adaptability to real-world industrial scheduling problems.

## 6 CONCLUSIONS

We propose the foundational models for industrial scheduling by leveraging techniques pivotal to the success of LLMs. Experimental results demonstrate that they exhibit significant effectiveness in addressing industrial scheduling problems. Our model achieves substantially superior performance and efficiency compared to Comparative algorithms, while also exhibiting scalability, generalization, and adaptability in industrial problems comparable to LLMs in NLP. Besides, we tailor techniques in LLMs to the specific characteristics of industrial scheduling problems and get better performance than directly using LLMs.

Furthermore, our foundational models transcend the conventional paradigm of developing specialized methods for individual scheduling problem types. This breakthrough enables the handling of complex real-world scheduling scenarios encountered involving diverse constraints, objectives, and operation types, due to using the flexible structure and fine-tuning techniques from LLMs.

Currently, there are growing skepticism in the AI community regarding the notion that "scaling LLMs is really the path to AGI?", with some even dismissing it as entirely unfounded. Our research demonstrates that even if it represent a detour on the path to AGI, the technologies and insights developed during the advancement of LLMs can still empower critical domains such as industrial and manufacturing sectors.

## 7 REPRODUCIBILITY STATEMENT

The code is available at `https://anonymous.4open.science/r/Foundation-Models-for-Industrial-Scheduling-7BD4`. It includes most of the content discussed in this paper, such as training, inference, and testing, except for portions involving proprietary corporate data. Besides, For the purpose of facilitating reader testing and replication, we distilled XXL(2B) to obtain a model "Distillate4Demo" ($\approx$ 60m) on $10 \times 10$ instances. Its performance is slightly worse than the original model, but still far better than other baseline algorithms. Even without the computational hardware (GPU: 8*H800 level), readers could run this distilled model ease.

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

## A    APPENDIX

### A.1    PPO USED IN OUR WORK

In this section, we briefly introduce the principles of the PPO algorithm employed in this study. In short, the goal of deep reinforcement learning is to find a policy $\pi$ that maximizes the expected return $J(\pi) = \int_\tau p(\tau \mid \pi)R(\tau) = \mathbb{E}_{\tau \sim \pi}[R(\tau)]$.

$$\pi^* = \arg\max_\pi J(\pi) = \arg\max_\pi \mathbb{E}_{\tau \sim \pi}[R(\tau)] \tag{A-1}$$

If $\pi$ is a neural network $\pi_\theta(A_t \mid S_t)$, we can use gradient ascent to optimize it. However, for complex problems where enumerating all possible states is infeasible, Monte Carlo methods are typically employed to estimate the expected return — that is, $J(\pi_\theta) \approx \frac{1}{|\mathcal{D}|} \sum_{\tau \in \mathcal{D}}[R(\tau)]$ — where the probability not explicitly shown can be used for gradient computation.

PPO uses an old policy $\pi_{\theta_{old}}$ to construct the implicit probability and calls it the surrogate objective.

$$
\begin{aligned}
\nabla_\theta J(\pi_\theta) &= \int_\tau \nabla_\theta p(\tau \mid \pi_\theta)R(\tau) \\
&= \int_\tau \frac{p(\tau \mid \pi_{\theta_{old}})}{p(\tau \mid \pi_{\theta_{old}})} \nabla_\theta p(\tau \mid \pi_\theta)R(\tau) \\
&= \int_\tau p(\tau \mid \pi_{\theta_{old}}) \frac{\nabla_\theta p(\tau \mid \pi_\theta)}{p(\tau \mid \pi_{\theta_{old}})}R(\tau) \\
&= E_{\tau \sim \pi_{\theta_{old}}}[\frac{\nabla_\theta p(\tau \mid \pi_\theta)}{p(\tau \mid \pi_{\theta_{old}})}R(\tau)] \\
&= E_{\tau \sim \pi_{\theta_{old}}}[\frac{\nabla_\theta \rho_0(S_0)\prod_{t=0}^T p(S_{t+1} \mid S_t, A_t)\pi_\theta(A_t \mid S_t)}{\rho_0(S_0)\prod_{t=0}^T p(S_{t+1} \mid S_t, A_t)\pi_{\theta_{old}}(A_t \mid S_t)}R(\tau)] \\
&= E_{\tau \sim \pi_{\theta_{old}}}[\frac{\nabla_\theta \prod_{t=0}^T \pi_\theta(A_t \mid S_t)}{\prod_{t=0}^T \pi_{\theta_{old}}(A_t \mid S_t)}R(\tau)] \\
\nabla_\theta J(\pi_\theta) &\approx E_{\tau \sim \pi_{\theta_{old}}}[\sum_{t=0}^T \frac{\nabla_\theta \pi_\theta(A_t \mid S_t)}{\pi_{\theta_{old}}(A_t \mid S_t)}R(\tau)]
\end{aligned}
\tag{A-2}
$$

Note that the last approximation sign holds only when $\pi_\theta(A_t \mid S_t) \approx \pi_{\theta_{old}}(A_t \mid S_t)$. Thus, PPO must constrain the divergence between the new and old policies within a certain range. PPO explores both KL divergence and direct clipping, with experimental results demonstrating that the latter is simpler and more efficient.

Let $r_t(\theta) = \frac{\pi_\theta(A_t|S_t)}{\pi_{\theta_{old}}(A_t|S_t)}$. The clipped surrogate objective is as follows.

$$\nabla_\theta J^{\text{CLIP}}(\pi_\theta) = E_{\tau \sim \pi_{\theta_{old}}} \sum_{t=0}^T \nabla_\theta \min[r_t(\theta)R(\tau), \text{clip}(r_t(\theta), 1-\epsilon, 1+\epsilon))R(\tau)] \tag{A-3}$$

To reduce variance, reinforcement learning often uses the advantage function, TD error, or generalized advantage estimator instead of $R(\tau)$. In order to reduce the coupling between actor and critic, our models choose to use the following simpler form:

$$\hat{R}_t = R(\tau_{t:}) - V_\phi(S_t) \tag{A-4}$$

where $V_\phi(S_t)$ is an estimate of the expected reward in state $S_t$. The loss function used for training is as follows:

$$L = [V_\phi(S_t) - R(\tau_{t:})]^2 \tag{A-5}$$

In order to balance exploration and exploitation, we also add an entropy. Therefore, the final critic loss function is as follows:

$$\nabla_\theta J^{\text{CLIP}}(\pi_\theta) = E_{\tau \sim \pi_{\theta_{old}}} \sum_{t=0}^{T} \nabla_\theta \min[r_t(\theta)\hat{R}_t,$$

$$\text{clip}(r_t(\theta), 1 - \epsilon, 1 + \epsilon))\hat{R}_t] \tag{A-6}$$

$$+ \beta \cdot \mathcal{H}(\pi_\theta(\cdot|S_t))$$

The PPO algorithm employed in our models is outlined in Algorithm A-1.

---

**Algorithm A-1** PPO Training

---

**Require:** Initial policy parameters $\theta_0$, initial value function parameters $\phi_0$

1: **for** $k = 0, 1, 2, \ldots, N$ **do**
2:     Collect set of trajectories $\mathcal{D}_k = \{\tau_i\}$ by running policy $\pi_{\theta_k}$ in the random FJSP
3:     Compute $\hat{R}_t$ based on the current value function $V_{\phi_k}$
4:     Fit value function according to Eq. A-5
5:     Update the policy according to Eq. A-6
6: **end for**
7: Return $\theta_N$ and $\phi_N$

---

The PPO hyperparameters used in our work are summarized in Table A-I. Notably, we implemented parallelized computation for the FJSP environment, enabling both the model and the environment to run efficiently on the GPU. This approach significantly reduces the communication overhead between the CPU and GPU.

Table A-I: Configuration of PPO during training.

| | | |
|---|---|---|
| Episodes | 80000 | |
| Batch size of envs | 32 | |
| Batch size of PPO | 1024 | |
| Reuse time | 1 | |
| $\epsilon$ | 0.1 | The range of clip |
| $\beta$ | 0.01 | The weight of the entropy to control the randomness of the policy |

Besides PPO, Group Relative Policy Optimization (GRPO) in DeepSeek-R1 is also a typical RL algorithm employed in LLMs. Based on PPO, GRPO abandons the value network and instead uses group relative rewards to compute the advantage function, thereby optimizing the policy model. By eliminating one network, GRPO significantly enhances stability during training, avoiding the coupling issues that arise from the simultaneous training of multiple networks. Therefore, we attempted to apply GRPO in our work, but the results were not as satisfactory as those obtained with PPO. We believe this is because the method of GRPO for estimating the value function.

Specifically, GRPO assumes a constant state value function estimation across all states within the same episode:

$$V(s) = V(s_0) = \overline{R},$$

where $\overline{R}$ denotes the mean return of trajectories sampled from the same initial state $s_0$. This approximation is acceptable for short-horizon tasks or single-decision problems, but it becomes problematic in long-horizon industrial scheduling settings.

When a trajectory yields a total return $R > \overline{R}$, GRPO uniformly reinforces all actions in that trajectory, irrespective of whether each individual action contributed positively or negatively to the overall

performance. Due to the Monte Carlo nature of trajectory sampling, each trajectory inevitably contains both favorable and unfavorable actions, which means that a large amount of poor actions would be reinforced. Moreover, as trajectory length increases, the variance of $R$ across trajectories diminishes, since the probabilities of beneficial and detrimental actions tend to be consistent under a fixed policy according to the Law of Large Numbers, causing most $R$ values to converge toward $\overline{R}$. This convergence leads to a substantial reduction in learning efficiency.

In contrast, PPO dynamically estimates $V(s_t)$ through a separate critic network, enabling the algorithm to discern beneficial from detrimental actions at each time step. This distinction allows PPO to reinforce advantageous behaviors while suppressing suboptimal ones, resulting in significantly more stable and efficient learning.

Although one could, in principle, modify GRPO to compute a separate group mean for each state $s_t$, this would require an exponentially growing number of groups as trajectory length increases, rendering the method computationally infeasible. Consequently, PPO remains the more practical and effective approach for long-horizon industrial scheduling tasks.

## A.2 HYPERPARAMETERS

Table B-I: Configuration of foundation models for industrial scheduling.

|       |              | XXXS    | XXS     | XS      | S       | N       | L       | XL      | XXL     |
|-------|--------------|---------|---------|---------|---------|---------|---------|---------|---------|
| Actor | dimension    | 64      | 256     | 256     | 512     | 512     | 512     | 1024    | 2048    |
|       | n layers     | 1       | 1       | 2       | 2       | 4       | 8       | 8       | 20      |
|       | n heads      | 8       | 8       | 8       | 16      | 16      | 16      | 16      | 32      |
|       | MLP ratio    | 4       | 4       | 4       | 4       | 4       | 4       | 4       | 4       |
|       | learing rate | 1.0E-04 | 1.0E-04 | 1.0E-04 | 1.0E-04 | 1.0E-05 | 1.0E-05 | 1.0E-05 | 3.0E-06 |
| Critic| dimension    | 64      | 256     | 256     | 512     | 512     | 512     | 1024    | 2048    |
|       | n layers     | 1       | 1       | 2       | 2       | 4       | 8       | 8       | 20      |
|       | n heads      | 8       | 8       | 8       | 16      | 16      | 16      | 16      | 32      |
|       | MLP ratio    | 4       | 4       | 4       | 4       | 4       | 4       | 4       | 4       |
|       | learing rate | 5.0E-05 | 5.0E-05 | 5.0E-05 | 5.0E-05 | 5.0E-06 | 5.0E-06 | 5.0E-06 | 1.5E-06 |
|       | total params | 0.1M    | 1.5M    | 3M      | 6M      | 25M     | 50M     | 200M    | 2B      |

In this section, we introduce hyperparameters of our models. In order to observe the scaling laws in the industrial scheduling base model, we trained a series of models with different scales. Referring to previous studies on scaling effects in LLMs, we trained and evaluated seven models from XXXS to XXL with the parameters shown in Table B-I.

As discussed in the main text, the instability in model training poses a significant challenge when scaling up the model size. We have attempted various approaches to mitigate this issue, such as implementing a backtracking or skipping mechanism to prevent loss spikes and employing GRPO to decouple the two networks. However, these methods demonstrated limited effectiveness. Consequently, we opted to reduce the learning rate proportionally with increasing model size. Additionally, to minimize experimental variables, we maintained a fixed learning rate throughout the training process. In accordance with established reinforcement learning practices, we set the critic's learning rate at half of the actor's learning rate.

The parameters for OR-tools used in this work are uniformly as follows: max_time_in_seconds = 36000; num_search_workers = 32; search_branching = AUTOMATIC; cp_model_presolve = True; cp_model_probing_level = 2; linearization_level = 0; absolute_gap_limit = 1e-6.

## A.3 THE DETAILS OF COMPARATIVE ALGORITHMS

### A.3.1 PRIORITY DISPATCHING RULES

Priority dispatching rules (PDRs) represent a classical approach for addressing FJSP, which repeatedly select the operation or machine with the highest priority based on some prescribed rules until a complete plan is generated. The FJSP can be decomposed into two interdependent sub-problems: (1) job sequencing and (2) machine selection. Consequently, solving FJSP requires two distinct types of dispatching rules. The job dispatching rule determines which operation from the pool of unscheduled operations should be processed next, while the machine selection rule identifies the most

suitable machine once an operation has been scheduled. We adopt four well-known job sequencing rules and two machine selection rules, combining them to form eight composite dispatching rules, with the top three performers serving as baseline methods in our main analysis.

**Job sequencing rules**:

1. FIFO (First in First Out): select the earliest arriving job in the queue of a machine.

2. MWKR (Most Work Remaining): select a job that has the most total processing time remaining to be done.

3. MOPNR (Most Operation Number Remaining): select a job that has the greatest number of remaining operations to be done.

4. LWKR (Least Work Remaining): select a job that has the least total processing time remaining to be done.

**Machine selection rules**:

1. EET (Earliest End Time): select a machine that is idle at the earliest time.

2. SPT (Shortest Processing Time): select a machine with the shortest processing time for the operation of a job.

In our experiments, FIFO+EET approach demonstrated the best overall performance. Consequently, in the main text, comparisons are primarily conducted using FIFO+EET.

### A.3.2 LEARNING-BASED ALGORITHMS

In addition to conventional PDRs, we evaluated our models against state-of-the-art learning-based approaches, including DRL Zhang et al. (2020) and MARL Lei et al. (2022). Here is the details:

1. DRL: The most classic deep reinforcement learning approach applied to JSSP. Utilizing graph neural networks as the core model, it demonstrates outstanding generalization capabilities at problem scale Zhang et al. (2020). We modified its input and output to enable matching with FJSP.

2. MARL: An end-to-end deep reinforcement framework to automatically learn two sub-policies, including a job operation action policy and a machine action policy to assign a job operation to a machine Lei et al. (2022).

### A.3.3 LLMS-BASED ALGORITHMS

We compared our approach with current LLM-based scheduling algorithms in order to verify whether the approach of employing LLMs in our model is more effective:

1. OPRO: Leveraging LLMs (GPT-4) as optimizers by iteratively refining solutions based on historical optimization trajectories provided in prompts Yang et al. (2024).

2. GEN: A fine-tuned LLM-based method (Meta-Llama-3.1-8B) that directly generates scheduling solutions in a text Abgaryan et al. (2024; 2025). Since the original method is designed for handling JSSP, we retrained and adjusted the prompt format to match FJSP.

### A.3.4 META-HEURISTIC ALGORITHMS

Although meta-heuristic algorithms and end-to-end algorithms differ significantly in principle, to comprehensively evaluate algorithm performance, particularly efficiency, we selected the following meta-heuristic algorithms for comparison:

1. 2SGA: A classic efficient two-stage GA to solve a comprehensive single-objective FJSP problem Defersha & Rooyani (2020).

2. HQPSO-VNS: A hybrid algorithm that integrates quantum particle swarm optimization and variable neighborhood search for efficiently addressing the FJSP Xu et al. (2024).

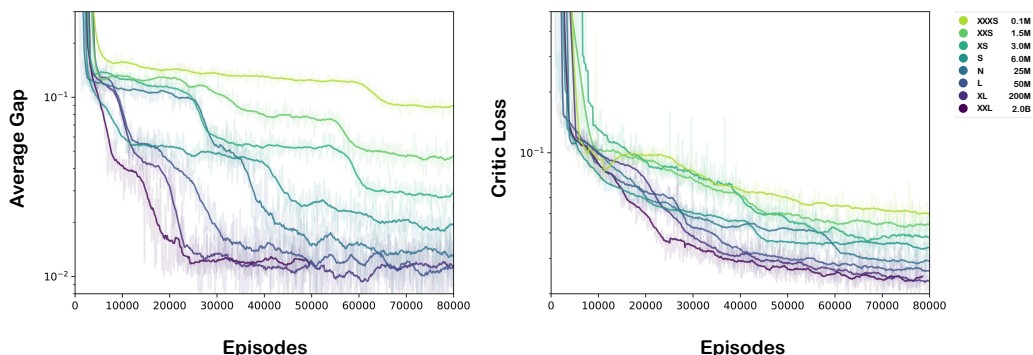

Figure D-1: The convergence curves of average gap and critic loss of our foundation models.

3. NSGA-II: A well-established multi-objective optimization algorithm that employs non-dominated sorting to obtain a set of non-dominated solutions (the Pareto front) Deb et al. (2002).

4. MOEA/D: An optimization approach designed to decompose multi-objective problems into simpler subproblems Zhang & Li (2007). For the FJSP, we further propose targeted improvements in population initialization and an adaptive neighborhood update mechanism Xiao et al. (2024).

To ensure consistent comparison, the population size and maximum iteration count are set to 400 for single-objective problems and 800 for multi-objective problems when applying the aforementioned algorithms.

## A.4 COMPLETE EXPERIMENT RESULTS

Table D-1, and D-3 presents detailed comparative results between our model and comparative algorithms (all PDRs, learning-based algorithms, LLMs-based algorithms and meta-heuristic algorithms) on the random dataset, Hurink's dataset Hurink et al. (1994) and real-world instances. Notably, our foundation models achieves marginally superior performance while demonstrating significantly faster computational speed (seconds versus hours) comparative to heuristic-based approaches.

Fig D-1 shows the convergence curves of average gap and critic loss of our foundation models. Due to the nature of RL, the oscillations of curves are far more intense than supervised learning. At the same time, except for the average gap, the loss of critic also decreases slightly with the increase of the model, which shows that the scaling law is also effective for critic. Using the same computational resources and parameters as in our work, training took around 12 hours to 300 hours (XXXS to XXL).

The problem scale may also influence model performance. To investigate this, we train and evaluate the models across varying problem scales by systematically adjusting $n$ and $l$. Since the absolute performance of the different models has already been presented in Fig. 5, our focus shifts to their relative performance trends as the problem scale increases. To quantify this, we use the average performance gap of models trained on the baseline configuration ($n = 10, l = 10$) as a reference point. The relative improvements of models trained on other data sizes are then computed and illustrated in Fig. D-2. The results indicate that larger models benefit more from the increased scale of training problems, but, consistent with previous findings, their performance ceases to improve beyond a certain model size.

Moreover, the performance gains from increasing the problem scale exhibit diminishing returns beyond a certain threshold. We hypothesize that this is because, for problem sizes exceeding a specific scale ($n \geq 15, l \geq 30$), the FJSP does not introduce additional complexity but rather becomes repetitive, leading to marginal differences in model performance. This observation also suggests that while larger models generally achieve superior results, they remain incapable of learning com-

Table D-1: Complete comparative results of the foundation model for industrial scheduling and comparative algorithms on randomly generated instances.

| Instance | | Dispatching rules | | | Learning-based | | LLMs-based | | Ours (XL) | $C_{max}^{BS}$ |
|---|---|---|---|---|---|---|---|---|---|---|
| | | FIFO + EET | MWKR+EET | MOPNR+EET | MARL | DRL | OPRO | GEN | | |
| 10x5 | Obj. | 674.46 | 693.96 | 696.17 | 576.16 | 574.15 | 611.20 | 565.79 | **554.19** | 545.75 |
| | Gap | 23.58% | 27.16% | 27.56% | 5.57% | 5.20% | 11.99% | 3.67% | **1.55%** | |
| | Time(s) | 0.08 | 0.09 | **0.08** | 0.10 | 0.15 | 45.00 | 1.04 | 0.12 | |
| 10x10 | Obj. | 694.42 | 701.66 | 684.85 | 635.92 | 634.80 | 732.61 | 635.47 | **633.92** | 630.27 |
| | Gap | 10.18% | 11.33% | 8.66% | 0.90% | 0.72% | 16.24% | 0.83% | **0.58%** | |
| | Time(s) | 0.14 | 0.16 | **0.14** | 0.30 | 0.47 | 103.00 | 1.95 | 0.19 | |
| 15x10 | Obj. | 929.32 | 897.75 | 915.01 | 834.6 | 829.86 | 1073.34 | 828.57 | **773.12** | 750.89 |
| | Gap | 23.76% | 19.56% | 21.86% | 11.15% | 10.52% | 42.94% | 10.35% | **2.96%** | |
| | Time(s) | 0.20 | 0.22 | **0.19** | 0.47 | 0.58 | 169.00 | 3.47 | 0.22 | |
| 20x10 | Obj. | 1097.89 | 1159.15 | 1082.05 | 1040.40 | 1035.53 | 1456.305 | 1041.79* | **1018.38** | 1009.51 |
| | Gap | 8.75% | 14.82% | 7.19% | 3.06% | 2.58% | 44.26% | 3.20% | **0.88%** | |
| | Time(s) | 0.28 | 0.3 | **0.26** | 0.61 | 0.77 | 245.00 | 4.18 | 0.58 | |
| 30x10 | Obj. | 1695.33 | 1640.50 | 1632.57 | 1549.77 | 1549.01 | 2265.4 | 1565.89* | **1527.04** | 1520.70 |
| | Gap | 11.48% | 7.88% | 7.36% | 1.91% | 1.86% | 48.97% | 2.97% | **0.42%** | |
| | Time(s) | 0.41 | 0.40 | **0.38** | 0.92 | 1.10 | 461.00 | 6.95 | 1.01 | |
| 40x10 | Obj. | 2150.74 | 2183.21 | 2133.54 | 2074.04 | 2057.45 | 3012.02 | 2079.74* | **2029.32** | 2017.94 |
| | Gap | 6.58% | 8.19% | 5.73% | 2.78% | 1.96% | 49.26% | 3.06% | **0.56%** | |
| | Time(s) | 0.55 | 0.59 | **0.51** | 1.22 | 1.45 | 548.00 | 8.17 | 1.37 | |
| 50x10 | Obj. | 2795.58 | 2848.06 | 2823.39 | 2575.57 | 2564.32 | 4264.13 | 2615.29* | **2536.65** | 2531.28 |
| | Gap | 10.44% | 12.51% | 11.54% | 1.75% | 1.31% | 68.46% | 3.32% | **0.21%** | |
| | Time(s) | 0.68 | 0.73 | **0.63** | 1.59 | 1.80 | 671.00 | 10.30 | 1.70 | |

Note: Obj. with "*" indicates that a legal solution could not be generated for part of the problems, i.e., the choice of machines and the number of operations did not match the requirements of the problems. In addition, since the LLMs-based OPRO is too time-consuming, we used the same method to calculate the average $C_{max}^{BS}$ to estimate it.

Table D-2: Comparative results of our foundation model for industrial scheduling and comparative algorithms on Hurink's instances.

| Instance | | Dispatching rules | | | Learning-based | | LLMs-based | | Ours (XL) | $C_{max}^{BS}$ |
|---|---|---|---|---|---|---|---|---|---|---|
| | | FIFO + EET | MWKR+EET | MOPNR+EET | MARL | DRL | OPRO | GEN | | |
| 10x5 | HurinkVdata4 | 640 (12.28%) | 644 (12.98%) | 672 (17.89%) | 610 (7.02%) | 610 (7.02%) | 618 (8.42%) | **570 (0.00%)** | 571 (0.18%) | 570 |
| | HurinkVdata5 | 624 (17.96%) | 613 (15.88%) | 653 (23.44%) | 555 (4.91%) | 554 (4.73%) | 594 (12.29%) | 542 (2.46%) | **537 (1.51%)** | 529 |
| | HurinkVdata6 | 572 (19.92%) | 566 (18.66%) | 558 (16.98%) | 532 (11.53%) | 528 (10.69%) | 565 (18.45%) | 524 (9.85%) | **484 (1.47%)** | 477 |
| | HurinkVdata7 | 649 (29.28%) | 593 (18.13%) | 605 (20.52%) | 530 (5.58%) | 528 (5.18%) | 549 (9.36%) | 528 (5.18%) | **509 (1.39%)** | 502 |
| | HurinkVdata8 | 582 (27.35%) | 577 (26.26%) | 522 (14.22%) | 507 (10.94%) | 510 (11.60%) | 540 (18.16%) | 501 (9.63%) | **462 (1.09%)** | 457 |
| 10x10 | HurinkVdata19 | 774 (7.95%) | 754 (5.16%) | 724 (0.98%) | 717 (0.00%) | 717 (0.00%) | 754 (5.16%) | 717 (0.00%) | **717 (0.00%)** | 717 |
| | HurinkVdata20 | 662 (2.48%) | 660 (2.17%) | 739 (14.40%) | 647 (0.15%) | 647 (0.15%) | 660 (2.17%) | 650 (0.62%) | **647 (0.15%)** | 646 |
| | HurinkVdata21 | 670 (1.06%) | 728 (9.80%) | 686 (3.47%) | 663 (0.00%) | 663 (0.00%) | 702 (5.88%) | 666 (0.45%) | **666 (0.45%)** | 663 |
| | HurinkVdata22 | 668 (8.27%) | 688 (11.51%) | 765 (23.99%) | 626 (1.46%) | 624 (1.13%) | 690 (11.83%) | 626 (1.46%) | **624 (1.13%)** | 617 |
| | HurinkVdata23 | 808 (6.88%) | 768 (1.59%) | 774 (2.38%) | 756 (0.00%) | 759 (0.40%) | 802 (6.08%) | 759 (0.40%) | **756 (0.00%)** | 756 |
| 15x10 | HurinkVdata24 | 929 (15.55%) | 935 (16.29%) | 997 (24.00%) | 887 (10.32%) | 885 (10.07%) | 1129 (40.42%) | 858 (6.72%) | **815 (1.37%)** | 804 |
| | HurinkVdata25 | 860 (16.85%) | 846 (14.95%) | 877 (19.16%) | 793 (7.74%) | 785 (6.66%) | 1067 (44.97%) | 775 (5.30%) | **747 (1.49%)** | 736 |
| | HurinkVdata26 | 878 (7.73%) | 905 (11.04%) | 961 (17.91%) | 858 (5.28%) | 854 (4.79%) | 1306 (60.25%) | 854 (4.79%) | **823 (0.98%)** | 815 |
| | HurinkVdata27 | 868 (12.00%) | 898 (15.87%) | 935 (20.65%) | 883 (13.94%) | 883 (13.94%) | 1094 (41.16%) | 837 (8.00%) | **801 (3.35%)** | 775 |
| | HurinkVdata28 | 937 (23.94%) | 930 (23.02%) | 914 (20.90%) | 883 (16.80%) | 878 (16.14%) | 1171 (54.89%) | 842 (11.38%) | **784 (3.70%)** | 756 |
| 20x10 | HurinkVdata29 | 1097 (4.08%) | 1159 (9.96%) | 1164 (10.44%) | 1089 (3.32%) | 1080 (2.47%) | 1579 (49.81%) | 1075 (1.99%) | **1064 (0.95%)** | 1054 |
| | HurinkVdata30 | 1195 (10.24%) | 1192 (9.96%) | 1219 (12.45%) | 1123 (3.60%) | 1115 (2.86%) | 1480 (36.53%) | 1123 (3.60%) | **1092 (0.74%)** | 1084 |
| | HurinkVdata31 | 1117 (4.39%) | 1212 (13.27%) | 1190 (11.21%) | 1106 (3.36%) | 1103 (3.08%) | 1485 (38.79%) | 1092 (2.06%) | **1084 (1.31%)** | 1070 |
| | HurinkVdata32 | 1063 (6.94%) | 1131 (13.78%) | 1140 (14.69%) | 1049 (5.53%) | 1046 (5.23%) | 1388 (39.64%) | 1050 (5.63%) | **1008 (1.41%)** | 994 |
| | HurinkVdata33 | 1127 (5.43%) | 1182 (10.57%) | 1164 (8.89%) | 1117 (4.49%) | 1120 (4.77%) | 1483 (38.73%) | 1116 (4.40%) | **1080 (1.03%)** | 1069 |
| 30x10 | HurinkVdata34 | 1577 (3.75%) | 1604 (5.53%) | 1632 (7.37%) | 1561 (2.70%) | 1561 (2.70%) | 1938 (27.50%) | 1567 (3.09%) | **1527 (0.46%)** | 1520 |
| | HurinkVdata35 | 1717 (3.56%) | 1774 (7.00%) | 1807 (8.99%) | 1693 (2.11%) | 1693 (2.11%) | 2165 (30.58%) | 1705 (2.83%) | **1680 (1.33%)** | 1658 |
| | HurinkVdata36 | 1556 (3.94%) | 1603 (7.08%) | 1599 (6.81%) | 1531 (2.27%) | 1527 (2.00%) | 1997 (33.40%) | N/A | **1506 (0.60%)** | 1497 |
| | HurinkVdata37 | 1620 (5.40%) | 1596 (3.84%) | 1611 (4.81%) | 1562 (1.63%) | 1561 (1.56%) | 2342 (52.37%) | 1569 (2.08%) | **1544 (0.46%)** | 1535 |
| | HurinkVdata38 | 1586 (2.39%) | 1665 (7.49%) | 1637 (5.68%) | 1574 (1.61%) | 1574 (1.61%) | 2239 (44.54%) | 1574 (1.61%) | **1557 (0.52%)** | 1549 |
| 40x10 | HurinkVdata44* | 2450 (5.92%) | 2483 (7.35%) | 2503 (8.21%) | 2353 (1.73%) | 2350 (1.60%) | 2931 (26.72%) | N/A | **2326 (0.56%)** | 2313' |
| | HurinkVdata45* | 2051 (5.83%) | 2044 (5.47%) | 2086 (7.64%) | 1972 (1.75%) | 1968 (1.55%) | 2647 (36.58%) | 2074 (7.02%) | **1945 (0.36%)** | 1938' |
| | HurinkVdata46* | 2306 (6.02%) | 2335 (7.36%) | 2345 (7.82%) | 2213 (1.75%) | 2213 (1.75%) | 3181 (46.25%) | N/A | **2188 (0.60%)** | 2175' |
| | HurinkVdata47* | 2063 (6.07%) | 2088 (7.35%) | 2105 (8.23%) | 1979 (1.75%) | 1976 (1.59%) | 2452 (26.07%) | 1968 (1.18%) | **1959 (0.72%)** | 1945' |
| | HurinkVdata48* | 2416 (5.96%) | 2439 (6.97%) | 2457 (7.76%) | 2320 (1.75%) | 2318 (1.67%) | 3391 (48.73%) | 2302 (0.96%) | **2288 (0.35%)** | 2280' |
| 50x10 | HurinkVdata49* | 2895 (5.50%) | 2935 (6.96%) | 2968 (8.16%) | 2782 (1.38%) | 2779 (1.28%) | 3745 (36.48%) | N/A | **2755 (0.40%)** | 2744' |
| | HurinkVdata50* | 2624 (5.59%) | 2661 (7.08%) | 2703 (8.77%) | 2528 (1.73%) | 2523 (1.53%) | 3488 (40.36%) | N/A | **2492 (0.28%)** | 2485' |
| | HurinkVdata51* | 2780 (5.82%) | 2756 (4.91%) | 2772 (5.52%) | 2663 (1.37%) | 2661 (1.29%) | 3642 (38.64%) | N/A | **2635 (0.30%)** | 2627' |
| | HurinkVdata52* | 2723 (8.06%) | 2699 (7.10%) | 2725 (8.13%) | 2558 (1.51%) | 2564 (1.75%) | 3335 (32.34%) | 2554 (1.35%) | **2532 (0.48%)** | 2520' |
| | HurinkVdata53* | 2625 (6.02%) | 2652 (7.11%) | 2704 (9.21%) | 2519 (1.74%) | 2519 (1.74%) | 3489 (40.91%) | N/A | **2485 (0.36%)** | 2520' |
| Ave. Gap | | 9.15% | 10.27% | 11.76% | 4.08% | 3.90% | 31.56% | 3.72% | 0.90% | |

Note: To observe the performance on larger scale problems, we generated two sets of test problems (40 × 10, 50 × 10) by combining small scale instances. "N/A" Indicates that the algorithm was unable to obtain a legal solution.

Table D-3: Comparative results of our foundation model for industrial scheduling, remaining PDRs and meta-heuristics algorithms on Hurink's instances.

| Instance | | Dispatching rules | | | | | Meta-heuristics (Avg) | | Ours | $C_{max}^{BS}$ |
|---|---|---|---|---|---|---|---|---|---|---|
| | | FIFO + SPT | MWKR+SPT | MOPNR+SPT | LWKR + SPT | LWKR + EET | HQPSO-VNS | 2SGA | | |
| 10x5 | HurinkVdata4 | 835 (46.49%) | 847 (48.60%) | 919 (61.23%) | 922 (61.75%) | 839 (47.19%) | 574.30 (0.75%) | 572.25 (0.39%) | **571 (0.18%)** | 570 |
| | HurinkVdata5 | 853 (61.25%) | 835 (57.84%) | 702 (32.70%) | 984 (86.01%) | 814 (53.88%) | 534.67 (1.07%) | **532.33 (0.63%)** | 537 (1.51%) | 529 |
| | HurinkVdata6 | 656 (37.53%) | 676 (41.72%) | 637 (33.54%) | 738 (54.72%) | 751 (57.44%) | 485.90 (1.87%) | **481.38 (0.92%)** | 484 (1.47%) | 477 |
| | HurinkVdata7 | 681 (35.66%) | 760 (51.39%) | 794 (58.17%) | 808 (60.96%) | 828 (64.94%) | 508.59 (1.31%) | **505.50 (0.70%)** | 509 (1.39%) | 502 |
| | HurinkVdata8 | 689 (50.77%) | 788 (72.43%) | 696 (52.30%) | 716 (56.67%) | 708 (54.92%) | 468.11 (2.43%) | 462.70 (1.25%) | **462 (1.09%)** | 457 |
| 10x10 | HurinkVdata19 | 1139 (58.86%) | 1067 (48.81%) | 1204 (67.92%) | 1241 (73.08%) | 1367 (90.66%) | 717.00 (0.00%) | 717.00 (0.00%) | **717 (0.00%)** | 717 |
| | HurinkVdata20 | 970 (50.15%) | 912 (41.18%) | 1044 (61.61%) | 1379 (113.47%) | 1305 (102.01%) | 649.05 (0.47%) | **646.00 (0.00%)** | 647 (0.15%) | 646 |
| | HurinkVdata21 | 1132 (70.74%) | 1053 (58.82%) | 983 (48.27%) | 1204 (81.60%) | 1280 (93.06%) | 666.82 (0.58%) | **663.00 (0.00%)** | 666 (0.45%) | 663 |
| | HurinkVdata22 | 978 (58.51%) | 999 (61.91%) | 1437 (132.90%) | 1188 (92.54%) | 1252 (102.92%) | 646.03 (4.71%) | **619.55 (0.41%)** | 624 (1.13%) | 617 |
| | HurinkVdata23 | 1029 (36.11%) | 1049 (38.76%) | 1022 (35.19%) | 1246 (64.81%) | 1596 (111.11%) | 756.00 (0.00%) | 756.00 (0.00%) | **756 (0.00%)** | 756 |
| 15x10 | HurinkVdata24 | 1403 (74.50%) | 1283 (59.58%) | 1292 (60.70%) | 1709 (112.56%) | 1941 (141.42%) | 857.83 (6.70%) | 825.58 (2.68%) | **815 (1.37%)** | 804 |
| | HurinkVdata25 | 1200 (63.04%) | 1350 (83.42%) | 1146 (55.71%) | 1527 (107.47%) | 1572 (113.59%) | 782.79 (6.36%) | 751.00 (2.04%) | **747 (1.49%)** | 736 |
| | HurinkVdata26 | 1287 (57.91%) | 1349 (65.52%) | 1271 (55.95%) | 1682 (106.38%) | 1770 (117.18%) | 856.19 (5.05%) | 834.00 (2.33%) | **823 (0.98%)** | 815 |
| | HurinkVdata27 | 1326 (71.10%) | 1378 (77.81%) | 1284 (65.68%) | 2096 (170.45%) | 1684 (117.29%) | 825.00 (6.45%) | **796.45 (2.77%)** | 801 (3.35%) | 775 |
| | HurinkVdata28 | 1370 (81.22%) | 1212 (60.32%) | 1393 (84.26%) | 1979 (161.77%) | 1547 (104.63%) | 801.67 (6.04%) | **777.55 (2.85%)** | 784 (3.70%) | 756 |
| 20x10 | HurinkVdata29 | 1611 (52.85%) | 1551 (47.15%) | 1393 (32.16%) | 2022 (91.84%) | 2087 (98.01%) | 1078.88 (2.36%) | 1064.55 (1.00%) | **1064 (0.95%)** | 1054 |
| | HurinkVdata30 | 1754 (61.81%) | 1630 (50.37%) | 1677 (54.70%) | 2401 (121.49%) | 2384 (119.93%) | 1106.65 (2.09%) | 1099.13 (1.40%) | **1092 (0.74%)** | 1084 |
| | HurinkVdata31 | 1596 (49.16%) | 1500 (40.19%) | 1693 (58.22%) | 2198 (105.42%) | 2366 (121.12%) | 1091.94 (2.05%) | 1091.00 (1.96%) | **1083.60 (1.27%)** | 1070 |
| | HurinkVdata32 | 1643 (65.29%) | 1533 (54.23%) | 1423 (43.16%) | 2240 (125.35%) | 2038 (105.03%) | 1018.81 (2.50%) | **1007.03 (1.31%)** | 1008 (1.41%) | 994 |
| | HurinkVdata33 | 1759 (64.55%) | 1545 (44.53%) | 1508 (41.07%) | 2267 (112.07%) | 1808 (69.13%) | 1094.21 (2.36%) | 1083.60 (1.37%) | **1080 (1.03%)** | 1069 |
| 30x10 | HurinkVdata34 | 2081 (36.91%) | 2141 (40.86%) | 2553 (67.96%) | 2525 (66.12%) | 2402 (58.03%) | 1529.35 (0.62%) | **1525.98 (0.39%)** | 1527 (0.46%) | 1520 |
| | HurinkVdata35 | 2277 (37.33%) | 2444 (47.41%) | 2454 (48.01%) | 2736 (65.02%) | 2839 (71.23%) | 1663.20 (0.31%) | 1665.78 (0.47%) | 1680 (1.33%) | 1658 |
| | HurinkVdata36 | 2305 (53.97%) | 2010 (34.27%) | 2198 (46.83%) | 2561 (71.08%) | 2587 (72.81%) | 1503.26 (0.42%) | 1504.28 (0.49%) | 1506 (0.60%) | 1497 |
| | HurinkVdata37 | 2248 (46.45%) | 2064 (34.46%) | 2242 (46.06%) | 2477 (61.37%) | 2717 (77.00%) | 1540.70 (0.37%) | 1540.80 (0.38%) | 1544 (0.59%) | 1535 |
| | HurinkVdata38 | 2098 (35.44%) | 2010 (29.76%) | 2174 (40.35%) | 2629 (69.72%) | 2653 (71.27%) | 1557.14 (0.53%) | 1555.85 (0.44%) | 1557 (0.52%) | 1549 |
| 40x10 | HurinkVdata44* | 3051 (31.91%) | 3182 (37.57%) | 3074 (32.90%) | 3800 (64.29%) | 3917 (69.35%) | 2339.50 (1.15%) | 2336.23 (1.00%) | **2326 (0.56%)** | 2313' |
| | HurinkVdata45* | 2746 (41.69%) | 2707 (39.68%) | 2732 (40.97%) | 3323 (71.47%) | 3386 (74.72%) | 1950.12 (0.63%) | 1951.49 (0.70%) | **1945 (0.36%)** | 1938' |
| | HurinkVdata46* | 2942 (35.26%) | 3029 (39.26%) | 2949 (35.59%) | 4043 (85.89%) | 4141 (90.39%) | 2194.66 (0.90%) | 2199.06 (1.11%) | **2188 (0.60%)** | 2175' |
| | HurinkVdata47* | 2663 (36.92%) | 2702 (38.92%) | 2683 (37.94%) | 3567 (83.39%) | 3695 (89.97%) | 1962.71 (0.91%) | **1958.20 (0.68%)** | 1959 (0.72%) | 1945' |
| | HurinkVdata48* | 3066 (34.47%) | 3098 (35.88%) | 3060 (34.21%) | 3507 (53.82%) | 3483 (52.76%) | 2296.73 (0.73%) | 2306.24 (1.15%) | **2288 (0.35%)** | 2280' |
| 50x10 | HurinkVdata49* | 3592 (30.90%) | 3604 (31.34%) | 3580 (30.47%) | 4145 (51.06%) | 4183 (52.44%) | 2760.77 (0.61%) | 2757.57 (0.49%) | **2755 (0.40%)** | 2744' |
| | HurinkVdata50* | 3233 (30.10%) | 3307 (33.08%) | 3218 (29.50%) | 3527 (41.93%) | 4226 (70.06%) | 2498.03 (0.52%) | 2492.67 (0.31%) | **2492 (0.28%)** | 2485' |
| | HurinkVdata51* | 3474 (32.24%) | 3444 (31.10%) | 3492 (32.93%) | 3669 (39.67%) | 4094 (55.84%) | 2641.82 (0.56%) | 2638.94 (0.45%) | **2635 (0.30%)** | 2627' |
| | HurinkVdata52* | 3291 (30.60%) | 3313 (31.47%) | 3320 (31.75%) | 3820 (51.59%) | 3919 (55.52%) | 2536.73 (0.66%) | **2530.46 (0.42%)** | 2532 (0.48%) | 2520' |
| | HurinkVdata53* | 3240 (30.86%) | 3303 (33.40%) | 3233 (30.57%) | 3914 (58.08%) | 4101 (65.63%) | 2487.25 (0.45%) | **2484.89 (0.36%)** | 2485 (0.36%) | 2520' |
| Ave. Gap | | 48.36% | 46.94% | 49.18% | 82.71% | 83.21% | 1.84% | 0.92% | 0.90% | |

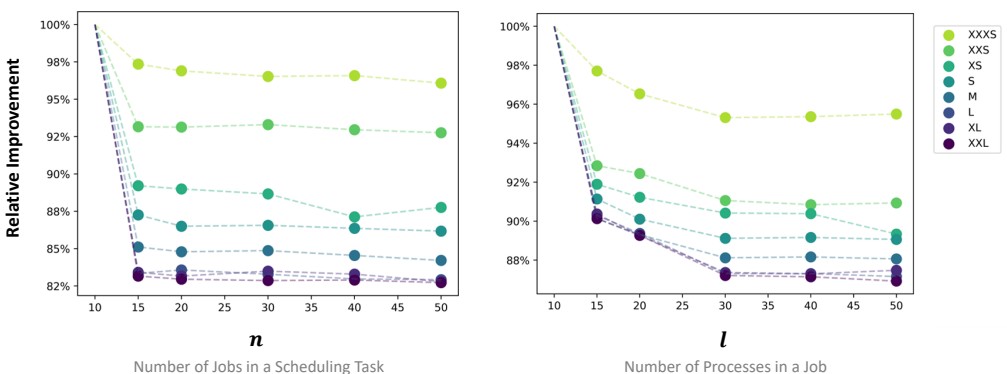

Figure D-2: Impact of problem scale on the performance of foundation model for industrial scheduling. To compare the trend of model performance with problem scale, we use the average gap of models trained on $n = 10, l = 10$ data as a baseline, and show the relative improvement of models trained on other data.

prehensive strategies solely from simple instances. To attain optimal performance, training must incorporate problem scales surpassing these thresholds.

Beyond the inherent complexity of the problem itself, training instability emerges as another critical obstacle that restricts the scalability and practical deployment of large models in industrial scheduling tasks. Among various instability phenomena, the loss spike is particularly detrimental, which refers to a sudden and sharp escalation in training error that often subsides rapidly within a few iterations. However, in our architecture, which integrates two interdependent networks, namely the actor and the critic, such transient spikes can cause reciprocal interference between the two components. Once this interference occurs, the networks may fail to stabilize and instead enter a progressive degradation cycle, ultimately preventing convergence, as illustrated in Fig. D-3. Furthermore, as the model capacity and parameter count increase, both the frequency and magnitude of loss spikes tend to intensify, leading to more frequent episodes of unstable learning and, in extreme cases, complete training collapse. We have

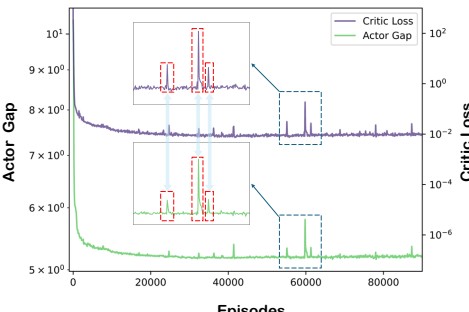

Figure D-3: The loss spikes during training, which are passed between the actor and critic, seriously affect the stability of training.

explored several remedial strategies, including rollback mechanisms and gradient skipping, yet these approaches provide only temporary relief. Through extensive experimentation, we find that lowering the learning rate remains the most reliable and effective mitigation for controlling instability and ensuring the model can resume smooth convergence.

## A.5 THE DETAILS OF REAL-WORLD INDUSTRIAL CASES

**Bulk Acoustic Wave (BAW) filter chips** are radio frequency devices that operate based on the piezoelectric effect, finding extensive applications in high-frequency domains including communication chips and advanced radar systems. The fundamental architecture of these filters comprises three essential components: a bottom electrode, a piezoelectric layer, and a top electrode. The majority of chip manufacturing processes remain consistent, showing characteristics of a non-permutation flow-shop scheduling problem. Moreover, variations in performance specifications and functional integration requirements lead to differences in the number of layers. Consequently, the number of repetitions of deposition and etching required is different for each batch of chips, as well as probe test and frequency modulation. It's a typical re-entrant scheduling problem. Furthermore, the time interval between deposition and etching must be strictly controlled to prevent oxidation and degradation of the coating. These characteristics are neglected in benchmark instances.

**Aircraft engine casings** are core components of an aircraft engine, primarily used to support and secure internal engine components while withstanding high temperatures, high pressure, and mechanical loads. Due to size constraints, neither rough machining nor finish machining of the casing can be completed in a single operation. The process must be divided into multiple sections. Each section requiring measurement upon completion to avoid wasteful machining of defective casings. There is no order to these operations, exhibiting characteristics of an open-shop scheduling problem. Besides, engine casings are typically manufactured and utilized in sets, hence quality consistency is also an optimization objective (achieving uniformity across equipment within the same set wherever possible).

The industrial data were collected from two Chinese manufacturers: a semiconductor fabrication facility in Sichuan and an aviation equipment producer in Beijing.

## A.6 LIMITATIONS

Here are the limitations of our work:

1. Scaling law exploration remains constrained by training instability and computational resources, with the largest model parameters currently reaching only the 2B scale, far below state-of-the-art LLMs. Future efforts should address training stability to validate scaling law at a larger scale.

2. Our foundation models are designed for practical industrial settings. Although we have established collaborations with actual factories, it has not yet been applied or validated in real industrial environments. This would be our future work.

