# OpenReview forum: "Foundation Models for Industrial Scheduling Leveraging the Techniques from LLMs"
_ICLR.cc/2026/Conference — Submitted to ICLR 2026_

### Official Review · Reviewer_F3mR · 2025-11-01

**Soundness:** 4
**Presentation:** 4
**Contribution:** 4
**Rating:** 8
**Confidence:** 4

**Summary:**

The Foundation Model for Industrial Scheduling is proposed to address the shortcomings of applying existing LLMs (both, general purpose and fine tuned for specific sched probs) directly to complex (mixed task types, multiobjective, contstraints) industrial scheduling problems, which often results in suboptimal, inefficient, or unreliable solutions due to the stringent requirements of real-world environments.

Through extensive experiments, the study demonstrates superior performance in solution quality and efficiency compared to conventional methods and algorithms that stiffly use LLMs for the tasks from the aerospace manufacturing field.

The evidence exhibits scaling law, generalization ability, and adaptability analogous to those observed in LLMs, may be providing the first empirical evidence that the scaling law also holds for industrial problems.

**Strengths:**

The experiments are quite comprehensive, covering comparisons across multiple algorithm categories, problem scales, industrial scheduling variants, and real-world industrial cases.
- Broad comparison against diverse algorithms (e.g., PDRs, RL, LLM based methods, metaheuristic algorithms)
- Validation Across Diverse Problem Scales and Instances (e.g., Randomly generated instances, standard benchmarks)
- FM scaling validation (e.g., various parameter sizes)
- Testing generalization and adaptability to variants (e.g., time limited FJSP, energy-aware multi-objective FJSP)
- Fine tuning techniques (e.g., the use of LoRA and concepts similar to ControlNet were validated to show how efficiently the FM could be adapted to these distinct objectives and constraints.
And at the end,
- Application to real world industrial cases from the aerospace manufacturing field

**Weaknesses:**

The potential weakness is the limited number of downstream tasks, which could limit the demonstration of its generalizability to industrial scheduling problems.
The model's adaptability to constraints was tested via time limits and the unique constraints of the BAW and engine casing production. For example, the current framework focuses on jobs and machines. Many industrial shops require limited shared resources (e.g., specialized tools, qualified personnel). Fine-tuning to include the scarcity and scheduling of a third resource type would be a complex test of the generalization capability of the architecture.

**Questions:**

- The paper notes that FJSP reduces to classical problems like the JSSP and the Non-permutation Flow-Shop Scheduling Problem by imposing constraints. Have the authors performed fine tuning experiments specifically targeting classical benchmark sets for these reduced problems (JSSP and Flow-Shop) to rigorously demonstrate that the general foundation is truly competitive with specialized, state-of-the-art solvers designed exclusively for those paradigms?
- The current fine-tuning focuses on makespan, time limits, and energy cost. Has the model been tested on problems where the primary objective is minimizing common industrial penalties, such as total or maximum job tardiness, or on problems that incorporate sequence-dependent setup times, which are ubiquitous constraints in manufacturing and are not explicitly detailed in the tested variants?

---

> ### Author Response · Authors · 2025-11-21
> **Responds to Reviewer 4**
>
> We sincerely thank the reviewer for their thoughtful and encouraging evaluation of our work. We truly appreciate the reviewer’s clear summary and positive assessment of our contribution, including the recognition of our comprehensive experimental validation, methodological soundness, and the paper’s presentation quality. We are grateful that the reviewer found our exploration of scaling laws, generalization, and real-world applicability to be strong aspects of the study.
>
> ### Respond to Questions
>
> #### Q1: *“The paper notes that FJSP reduces to classical problems like the JSSP and the Non-permutation Flow-Shop Scheduling Problem by imposing constraints. Have the authors performed fine tuning experiments ...? ”*
>
> We appreciate the reviewer’s insightful comment. Indeed, we have conducted experiments applying our framework to other classical scheduling problems, including the JSSP, Flow-Shop, and Open-Shop variants. However, due to time constraints, we were unable to collect and organize a comprehensive set of results across all baseline algorithms at the time of submission, and thus these were not included in the original version. In the revised manuscript, we have now added these experimental results in **Fig. 8**, which consistently demonstrate that our model maintains strong performance across different types of scheduling problems. We also plan to further extend the comparisons with additional state-of-the-art algorithms in future work to provide a more complete evaluation.
>
> #### Q2: *“The current fine-tuning focuses on makespan, time limits, and energy cost. Has the model been tested ...?”*
>
> We appreciate the reviewer’s valuable suggestion. We fully agree that objectives such as total or maximum job tardiness and sequence-dependent setup times are of higher practical importance in industrial settings than simple makespan minimization. Due to time limitations, particularly the need to retrain comparison algorithms in addition to fine-tuning our model, we were unable to include these experiments in the current submission. However, we believe adapting our framework to such objectives would not be difficult, as our experiments already involve multi-objective formulations where the model demonstrates strong performance even under more complex optimization goals.
>
> Moreover, For problems that incorporate sequence-dependent setup times, we have already considered similar constraints in some industrial cases, for example, the time-interval restrictions between etching and deposition steps in semiconductor manufacturing. We plan to extend our study to explicitly include tardiness and sequence-dependent setup time objectives in future work.
>
> ### Responds to Weaknesses
>
> We sincerely thank the reviewer for the thoughtful and constructive comments. We fully agree that the mentioned aspects represent important directions for extending our work. These issues, such as incorporating additional resource types and testing broader downstream tasks, will be key focuses in our future research and model development.

---

### Official Review · Reviewer_F3y3 · 2025-11-01

**Soundness:** 3
**Presentation:** 3
**Contribution:** 3
**Rating:** 4
**Confidence:** 3

**Summary:**

The paper introduces a novel approach: the construction of a general-purpose **Foundation Model for Industrial Scheduling (FMIS)** by drawing upon key techniques from Large Language Models (**LLMs**), such as the **Decoder-only architecture** and **Scaling Law**.
The core objective is to address the over-specialization of existing scheduling methods for specific problem types, while simultaneously overcoming the challenges of **reliability, data scarcity, and specialization** that arise when applying LLMs directly to industrial settings.

**Strengths:**

1.  The core idea—leveraging the **"technology stack" of LLMs** (e.g., Decoder-only architecture, Scaling Law) rather than directly applying LLMs themselves to solve industrial scheduling problems—is highly innovative. This methodological shift effectively bypasses challenges like reliability, data sparsity, and specialization encountered when deploying LLMs in industrial settings, offering a new pathway for AI implementation in the industry.
2.  By modeling the Flexible Job Shop Scheduling Problem (FJSP) as the core problem and utilizing LLM-inspired design principles, the model demonstrates the potential to handle a mix of task types, objectives, and constraints. Experimental results, especially across various scales of FJSP instances, constrained variants, and real-world industrial cases, strongly support FMIS's superiority in generality and generalization over traditional specialized methods.
3.  **First Empirical Evidence of Scaling Law**: This paper provides the first empirical evidence of the **Scaling Law** for industrial scheduling problems (though limitations exist at extremely large scales). This not only offers theoretical guidance for future industrial AI model development but also broadens our understanding of the "large model" phenomenon, suggesting its underlying mechanisms may extend beyond Natural Language Processing.

**Weaknesses:**

1.   While PPO is common in LLM alignment, the paper primarily relies on PPO for training, which differs from the "Pre-training + RLHF" paradigm of LLMs. The authors mention the "inherent limitations of industrial datasets, hence primarily adopting RL." Does this imply that scheduling tasks inherently lacking large-scale pre-training data are better suited for RL, or is the pre-training capacity of LLMs hard to transfer to the discrete scheduling action space? This point requires further clarification. Furthermore, the explanation for the failure of GRPO (inaccurate intermediate state value function estimation) warrants deeper investigation.
2. Although a Decoder-only and self-attention mechanism is used, the model does not directly process "token sequences" or "natural language instructions" like LLMs. The input encoding (Job State, Machine State, Operation) and output (probability matrix $P_{(n,l) \times m}$) are highly structured. This makes its similarity to LLMs lie more in the underlying Transformer structure and the Scaling Law, rather than in language understanding or generation capabilities. While not necessarily a drawback, the authors should more clearly delineate its difference from traditional Transformer-based methods and the specific meaning of "Foundation Model" in the industrial scheduling context.
3.   The paper defines "Foundation Model" as a general framework trained on FJSP to adapt to various scheduling problems. However, while FJSP is general, its task definition and data structure are relatively fixed. It remains unclear how effectively the model can be applied to optimization problems with completely different structures (e.g., resource allocation, routing problems), and what degree of fine-tuning or architectural adjustment would be necessary. This impacts the breadth that the term "Foundation Model" can truly encompass in the industrial domain.

**Questions:**

1.   You mention that model performance plateaus after 500M parameters. Do you believe this is due to an inherent complexity limit of industrial scheduling problems, suggesting a performance ceiling for the optimal solution, or is it a limitation of the current model architecture or RL training paradigm? Is there potential to break this bottleneck by improving the architecture or training strategy in the future?
2.  How is $F(S)$ in the formula $P_{(n,l) \times m} = \text{Softmax}(\text{Linear}(x) - F(S) \cdot \infty)$ specifically constructed? For instance, how are dynamic and complex rules such as "machine failure" or "task cancellation" encoded into the $F(S)$ matrix? What is the impact of this rule integration method on the model's scalability and real-time performance?
3.  You mention that GRPO failed to achieve satisfactory results, partly due to inaccurate value estimation, especially for long-horizon trajectories. This implies a challenge posed by the sequence length and complexity of industrial scheduling problems to RL algorithms. Have you considered other RL algorithms, such as tree-search-based RL methods (e.g., AlphaZero) or Offline RL, which might have advantages in handling long trajectories and leveraging limited data?
4.  The paper highlights the limitations of industrial datasets. Although RL is used for training, the success of LLMs hinges on massive pre-training. Is there a possibility for a "general scheduling corpus" for large-scale pre-training in the industrial scheduling domain? Or does the "Foundation" of FMIS refer more to its structure rather than "general knowledge" acquired through vast data, as with LLMs?
5.  In discussing limitations, you mention that the model has not yet been applied or validated in real industrial environments. Given the strict requirements of industrial settings, what are the main challenges you foresee in deploying FMIS from a laboratory setting to an actual factory? For example, how to handle real-time data streams, ensure decision interpretability, and integrate with existing Manufacturing Execution Systems (MES)?

---

> ### Author Response · Authors · 2025-11-21
> **Respond to Questions**
>
> We sincerely thank the reviewer for the constructive and insightful feedback. We are especially grateful for your recognition of our work’s contribution in providing the *first empirical evidence of the Scaling Law in industrial scheduling problems*. Validating this phenomenon required extensive experiments and significant computational and engineering effort, so your acknowledgment is deeply encouraging. We truly appreciate your thoughtful comments and the recognition of our methodological innovation and empirical rigor.
>
> #### Q1: *“You mention that model performance plateaus after 500M parameters. Do you believe this is due to an inherent complexity ...? ”*
>
> Thank you for this insightful question. We believe the performance plateau around 500M parameters primarily arises from three factors:
>
>  (1) **Problem complexity:** Although industrial scheduling tasks can be infinitely varied through parameter changes, they still belong to a single problem class and cannot match the nearly unbounded data diversity available to LLMs.
>  (2) **Reward sparsity:** The discrete nature of scheduling solutions causes the reward distribution to become highly sparse as performance improves, making it increasingly difficult for Monte Carlo sampling to discover better solutions.
>  (3) **Model instability:** As model size increases, training becomes progressively unstable.
>
> In our experiments, factors (1) and (2) dominate, while (3) becomes critical only when model size exceeds ~8B parameters, where training becomes unstable. To overcome this bottleneck, future work may focus on **increasing problem diversity and complexity**, and **reducing interference between the actor and critic networks** to achieve more stable and scalable learning on harder scheduling tasks.
>
> #### Q2: *“How is $F(s)$ in the formula ...?”*
>
> Thank you for the question. The function $F(s)$ mainly acts as a **simple yet effective trick**: it masks infeasible actions—e.g., machines under maintenance, unfinished predecessors, or operations violating time constraints—by setting their output probabilities to zero. Therefore, $F(\cdot)$ is a simple function based on fundamental rules of industrial scheduling. Though conceptually straightforward, this greatly **enhances training stability and speed** with **almost no extra computational cost**, making it especially helpful for large-scale models.
>
> #### Q3: *“You mention that GRPO failed to achieve ...?”*
>
> We appreciate this insightful suggestion. Tree-search-based RL methods such as those used in AlphaGo are indeed appealing, and we have explored this idea. However, tree search is much more computationally intensive than simple Monte Carlo sampling, which can be efficiently parallelized (8×H800 GPUs, batch size up to 4096). In contrast, tree search depends on previous states, making full parallelization difficult and greatly reducing efficiency. For better scalability, we therefore adopted the latter. Nevertheless, your suggestion is inspiring, using tree-search-based optimization *after* model pre-training may further enhance performance. We would check this in the future work.
>
> #### Q4: *“The paper highlights the limitations ...?”*
>
> Thank you for the thoughtful question. Building a large-scale “industrial scheduling corpus” is indeed an appealing idea. However, unlike natural language data that naturally aggregates on the internet, industrial data are **scattered across individual factories**, often **undigitized or inaccessible** due to limited information systems. This has also been a major challenge in our industrial collaborations. Therefore, our approach focuses on **training a foundation model within a unified simulation environment**, which can then be **fine-tuned for specific factory scenarios**. In this sense, the “foundation” of FMIS refers to its **structural generality and adaptability**, rather than large-scale pretraining on real data.
>
> #### Q5: *“In discussing limitations, you mention that the model has not yet ...?”*
>
> Thank you for this valuable question. Our work is currently undergoing preliminary validation in a real factory setting. The main challenge we face lies in **fragmented information flows**—different machines and vendors use incompatible software and data formats, making system integration the biggest obstacle. Incomplete digital infrastructure thus remains a key barrier to industrial intelligence. Regarding interpretability, since scheduling is essentially an **optimization problem**, verifying a good solution is far easier than discovering one. Therefore, in practice, **solution quality** matters more than model interpretability, which is less critical in this context.

---

> ### Author Response · Authors · 2025-11-21
> **Responds to Weaknesses**
>
> We fully agree with the insightful points raised by the reviewer. The issues you highlighted have provided crucial directions for improving our study. We sincerely thank you for your constructive review, which has been instrumental in enhancing the academic depth and rigor of this paper. Below are our specific discussions.
>
> #### 1. Respond to Weakness 1
>
> Thank you for the insightful comments.
>
> For the first question, we agree that, due to the lack of large-scale data, a self-supervised pre-training strategy like that used in LLMs is not suitable here. Our goal is not only automation but also **surpassing existing industrial scheduling algorithms**, which cannot be achieved by merely learning from current data (similar to how AlphaZero abandoned human-game imitation in favor of total self-play). Besides, since industrial systems are human-designed, they are more in control and easier to simulate. A virtual industrial environment naturally fits **RL**.
>
> Regarding the **GRPO**, we will expand our explanation in the revised paper. The main issue lies in the way GRPO estimates the value function. Specifically, GRPO assumes a constant state value estimation for all states within the same episode:
> $$
> V(s_t) = V(s_0) = \overline{R}
> $$
> where $\overline{R}$ is the mean return of trajectories sampled from the same initial state $s_0$. This approach works for **short-horizon tasks** or **single-decision problems**, but in industrial scheduling with a long trajectory it becomes problematic.
>
> If a trajectory achieves a total return $R > \overline{R}$, all actions within that trajectory are equally encouraged, **regardless of whether each individual action contributed positively or negatively** to the final outcome. Due to the Monte Carlo nature of trajectory sampling, each trajectory inevitably contains both good and bad actions. Besides, as trajectories grow longer, the variance of $R$ across trajectories decreases (since under the same policy, the probability of a good action and a bad action is consistent & Law of Large Numbers), causing most $R$ values to approach $\overline{R}$. This leads to severely **degraded learning efficiency**.
>
> [Diagram illustrating the reasons for GRPO's suboptimal performance](https://postimg.cc/14QzqTPz)
>
> In contrast, **PPO** estimates $V(s_t)$ dynamically using a separate critic network. This allows the algorithm to distinguish between beneficial and detrimental actions at each time step, reinforcing good actions while suppressing bad ones, leading to far more stable and efficient learning.
>
> In principle, one could modify GRPO to compute a separate group mean for each state $s_t$, but this would require exponentially increasing the number of groups with trajectory length, making it **computationally intractable**. Hence, PPO remains the more practical and effective choice for long-horizon industrial scheduling tasks.
>
> #### 2. Respond to Weakness 2
>
> Thank you for this helpful comment. We agree that our model adopts only the **underlying architectural principles of LLMs** (e.g., Decoder-only Transformer and Scaling Law), while its operation mode differs. Specifically, we use a **state-to-action** formulation rather than the **sequence-to-next-token** prediction used in language modeling, as this design better fits the characteristics of industrial scheduling. To clarify this distinction, we have **revised the Methods section** and updated **Fig. 2 and Fig. 3** to make the differences more explicit to readers.
>
> Regarding the use of the term **“Foundation Model”**, we agree that it can be ambiguous, which in many contexts, it is often directly equated with LLMs. In our paper, we use the term in its **original sense**—a model trained via **pre-training and fine-tuning** to achieve broad adaptability across tasks, rather than as synonymous with LLMs trained on massive general-purpose corpora. Our FMIS is pre-trained on a **general FJSP environment** to learn broadly applicable scheduling principles and then **fine-tuned for specific industrial scenarios**, thus embodying the foundational, adaptable nature of the term in the industrial AI context.
>
> #### 3. Respond to Weakness 3
>
> Thank you for the thoughtful comment. Our proposed FMIS is indeed designed as a **foundation model for industrial scheduling problems**. Given the wide diversity of problem types in industry, building a single model that efficiently handles all of them remains highly challenging. However, problems such as **resource allocation** and **routing** are essentially forms of **combinatorial optimization**. Thanks to the strong flexibility of the **decoder-only architecture** and **attention mechanism**, our approach can be readily adapted to these tasks by **redefining the input and output representations**. We believe this offers a practical pathway toward broader generalization in future work.

---

### Official Review · Reviewer_1cih · 2025-11-01

**Soundness:** 2
**Presentation:** 3
**Contribution:** 2
**Rating:** 4
**Confidence:** 4

**Summary:**

The paper proposes foundation models for industrial scheduling by adapting LLM techniques (decoder-only architecture, scaling laws) to flexible job-shop scheduling problems (FJSP). It claims these models generalize across industrial variants (time-limited, energy-aware) through fine-tuning. While demonstrating scaling laws and outperforming baselines (DRL, LLM-based OPRO/GEN), the work overlooks critical structural dependencies in scheduling DAGs, limiting its industrial applicability.

**Strengths:**

__Originality__: The paper creatively bridges LLM advancements and industrial scheduling, moving beyond prior single-problem approaches (Xiong et al., 2022b) to propose a unified foundation model framework. Its adaptation of scaling laws to FJSP (Fig. D-1/D-2) offers a novel perspective for industrial AI.

__Significance__: Addressing the gap between academic scheduling benchmarks and real-world industrial complexity (e.g., re-entrant processes, time-limited operations could impact manufacturing efficiency. The LoRA fine-tuning approach for multi-objective variants shows practical potential

**Weaknesses:**

__Missing DAG Context__: The decoder-only architecture processes operations sequentially without modeling job dependencies as a DAG. Industrial scheduling requires respecting precedence constraints (Fig. 1), but the model treats operations as isolated tokens ("O1,1", "O2,1"), ignoring temporal dependencies critical for feasible schedules.

__Overlooked Industrial Realities__: The paper acknowledges re-entrant scheduling and oxidation constraints (section 3), yet the model's "modified causal masking" only filters operations without encoding dependency graphs. Real-world examples (e.g., chip manufacturing requiring strict deposition-etching intervals) demand explicit DAG representation.

__LLM Adaptation Limitations__: Directly borrowing decoder-only designs neglects scheduling's combinatorial nature. Unlike text generation, scheduling requires global constraint satisfaction—evident in Fig. D-3's training instability from unmodeled dependencies.

**Questions:**

1. How would your model represent scheduling constraints like "operation A must precede B with ≤5min interval" in the DAG structure? Current token-based inputs (section 4.3) seem insufficient for temporal dependencies.
2. Why prioritize decoder-only architectures over graph-based models (e.g., GNNs) that natively capture job dependencies? Could this choice explain the critic loss spikes in Fig. D-3?


__MISSING__ code link, the JSSP benchmark is quite limited.

---

> ### Author Response · Authors · 2025-11-16
> **For MISSING code link**
>
> I must apologise for failing to notice that the anonymous link had expired due to an oversight. I have now updated the link.
>
> You may now access my code via the link in the paper or here [https://anonymous.4open.science/r/Foundation-Models-for-Industrial-Scheduling-7BD4](https://anonymous.4open.science/r/Foundation-Models-for-Industrial-Scheduling-7BD4).

---

> ### Author Response · Authors · 2025-11-21
> **Respond to Question**
>
> We sincerely thank the reviewer for their thoughtful and constructive feedback, as well as for recognizing the originality and potential significance of our work. We are particularly grateful for the reviewer’s acknowledgment of our effort to bridge LLM advancements with industrial scheduling.
>
> ### Respond to Questions
>
> #### Q1: *''How would your model represent scheduling constraints like "operation A must precede B with ≤5min interval" in the DAG structure? Current token-based inputs (section 4.3) seem insufficient for temporal dependencies.''*
>
> We agree that fully modeling temporal dependencies among multiple operations would ideally require a DAG-based representation. However, the temporal constraints considered in our work are local, i.e., operation *A* must be executed within a specified interval after a preceding operation *B* (as in chip manufacturing where etching steps require a controlled time gap). In such cases, explicit graph construction is unnecessary since the dependency scope is fixed. We incorporate this constraint by augmenting the operation token with timing information:
>
> $$
> O^*_{i,k_a} = O_{i,k_a} + \text{Embedding}(k_b, T)
> $$
>
> where $k_a, k_b$ denote the indices of operations *A* and *B*, and $T$ is the required interval. $O_{i,k}$ is the encoded token of operation $k$ for job $i$, and $O^*_{i,k}$ represents the constraint-enhanced token. This allows the model to learn local temporal dependencies while maintaining the simplicity and scalability of the token-based input design.
>
> #### Q2: Why prioritize decoder-only architectures over graph-based models (e.g., GNNs) that natively capture job dependencies? Could this choice explain the critic loss spikes in Fig. D-3?
>
> We chose a decoder-only architecture primarily because it has been extensively validated in large-scale LLM research, where model parameters can scale to hundreds of billions — a prerequisite for reliably studying *scaling laws*. In contrast, current investigations of scaling behavior in GNNs remain preliminary, with parameter counts typically below 100M (often ≤10M), as reported in **Liu et al. (2024)** and **Wang et al. (2024)**. To rigorously test scaling effects in industrial scheduling — one of our paper’s central contributions — we therefore adopted a decoder-only model that naturally supports billion-scale training.
>
> - Liu, Jingzhe, et al. *“Towards Neural Scaling Laws on Graphs.”* *The Third Learning on Graphs Conference (LoG)*, 2024.
> - Wang, Zhen, et al. *“Exploring Neural Scaling Law and Data Pruning Methods for Node Classification on Large-Scale Graphs.”* *Proceedings of the ACM Web Conference (WWW)*, 2024.
>
> Regarding the *critic loss spikes* in Fig. D-3, we note that such instabilities are a known phenomenon in large-scale model optimization and have been theoretically analyzed (e.g., **Molybog et al., 2023**; **Li et al., 2023**). These loss spikes typically emerge as model size grows, independent of architecture type. Therefore, we attribute the observed instability to large-model training dynamics rather than to the decoder-only design itself.
>
> - Molybog, Igor, et al. *“A Theory on Adam Instability in Large-Scale Machine Learning.”* *arXiv preprint arXiv:2304.09871*, 2023.
> - Li, Xiaolong, Zhi-Qin John Xu, and Zhongwang Zhang. *“Loss Spike in Training Neural Networks.”* *arXiv preprint arXiv:2305.12133*, 2023.
>
> #### Q3: Missing Code
>
> We apologize for the broken code link. The anonymous repository expired during the review process due to our oversight. This issue has now been resolved, and the code is again accessible via this anonymous link.
>
> #### Q4: The JSSP benchmark is quite limited.
>
> We appreciate the reviewer’s comment regarding the bencshmark scope. Our foundation model demonstrates strong generalization across scheduling variants; we have therefore added results on standard **JSSP benchmarks** ( Taillard’s instances ) , as well as on **NPFSP** and **OJSP**. These new results, now included in the revised version in **Fig. 8**, further confirm the adaptability of our approach.

---

> ### Author Response · Authors · 2025-11-21
> **Responds to Weakness**
>
> We fully agree with the insightful points raised by the reviewer. The issues you highlighted have provided crucial directions for improving our study. We sincerely thank you for your constructive review, which has been instrumental in enhancing the academic depth and rigor of this paper. Below are our specific discussions.
>
> #### 1. Missing DAG Context
>
> We agree that a DAG is essential for fully representing operation dependencies in industrial scheduling. In our current work, dependencies are encoded through re-usable operation indices, which can capture specific but limited dependency patterns.
>
> For example, our models would support the following dependency,
> $$
> \begin{matrix}
>    &   & O_3 &   &    \\
>    &\nearrow && \searrow & \\
>   O_1\to O_2 &&&& O_4 &\to & O_5 \\
>   &\searrow && \nearrow & \\
>   &   & O_3 &   &    \\
> \end{matrix}
> $$
> but don't support the the following dependency,
> $$
> \begin{matrix}
>    &   &  &O_3&&   &    \\
>    &\nearrow &&&& \searrow & \\
>   O_1\to O_2 &&&&&& O_4 &\to & O_5 \\
>   &\searrow &&&& \nearrow & \\
>   &   & O_6 &\to& O_7&   &    \\
> \end{matrix}
> $$
>
> [Alternative image to prevent incorrect display of formula: Examples our model can support and cannot support](https://postimg.cc/gxCd8MQL)
>
> As the reviewer correctly points out, embedding dependency information directly into tokens cannot, in principle, represent an arbitrary number of incoming edges in a DAG, since the encoding length must remain finite.
>
> We truly appreciate the reviewer’s insightful comment, achieving full DAG compatibility is indeed a key step toward a comprehensive *foundation model for industrial scheduling*. Although our current model cannot express arbitrary DAGs, in practical industrial scenarios the number of predecessors per operation is typically bounded. Inspired by your suggestion, we plan to introduce a fixed-length representation for multiple predecessors, enabling **partial DAG support** within our tokenization scheme. While we could not include this extension in the current version due to time constraints, it has been added to our project roadmap and will be implemented in follow-up work.
>
> #### 2. Overlooked Industrial Realities
>
> We agree that incorporating a full DAG representation is essential for complex industrial scenarios. As noted in our response above, we have already included partial DAG support in our project plan. The *re-entrant scheduling* and *oxidation constraints* considered in our current work are relatively localized — for example, in chip manufacturing, etching steps are only constrained by preceding deposition processes, without broader inter-operation dependencies. Therefore, our current method already satisfies these simpler cases. We sincerely appreciate this valuable suggestion, which will guide our future efforts toward a truly general foundation model for industrial scheduling.
>
> #### 3. LLM Adaptation Limitations
>
> We agree that modeling global dependencies—especially through a DAG representation—is crucial for capturing the full structure of scheduling problems. However, our approach does not simply apply a standard decoder-only architecture without adaptation. To better align with scheduling characteristics, we replace the *sequence-to-next-token* paradigm of LLMs with a *state-to-action* formulation, enabling the model to learn decision-making dynamics specific to scheduling tasks.
>
> The instability observed in **Fig. D-3** is not caused by “unmodeled dependencies.” This phenomenon only emerges when the model scales beyond **1B parameters**, whereas models below **100M parameters** (still far larger than typical GNN or DRL-based schedulers) train stably. Both our experiments and prior studies on large-model optimization show that *loss spikes become more frequent as parameter scale increases*, independent of dependency modeling. **If the instability in training stems from “unmodeled dependencies”, it would be impossible to explain why the instability occurs only when the model is larger.**   This is because all models of different scales employ the same approach to problem modelling.

---

### Official Review · Reviewer_fmji · 2025-11-01

**Soundness:** 4
**Presentation:** 4
**Contribution:** 4
**Rating:** 8
**Confidence:** 4

**Summary:**

The paper proposes an original decoder only architecture for scheduling problems and compares it to state of the art algorithms and LLMs. It has good results on standard and industrial benchmarks even for multi-objective problems.

**Strengths:**

The paper proposes an original architecture for scheduling problems that outperforms other approaches.
The paper is clearly presented
The addressed problem is an important problem
The proposed framework is general

**Weaknesses:**

The code is unavailable
The mention to AGI in the conclusion is superfluous

**Questions:**

How does your method compare to usual Operational Research algorithms such as MIP?

---

> ### Author Response · Authors · 2025-11-16
> **For MISSING code link**
>
> I must apologise for failing to notice that the anonymous link had expired due to an oversight. I have now updated the link.
>
> You may now access my code via the link in the paper or here [https://anonymous.4open.science/r/Foundation-Models-for-Industrial-Scheduling-7BD4](https://anonymous.4open.science/r/Foundation-Models-for-Industrial-Scheduling-7BD4).

---

> ### Author Response · Authors · 2025-11-21
> **Response to Reviewer 1**
>
> We sincerely thank the reviewer for the positive evaluation of our work and for recognizing its originality, clarity, and contributions.
>
> We address the raised concerns below.
>
> #### 1. Code availability:
>
> We apologize for the broken code link. The anonymous repository expired during the review process due to our oversight. This issue has now been resolved, and the code is again accessible via the anonymous link.
>
> #### 2. Mention of AGI:
>
> We agree that the reference to AGI was unnecessary. These mentions have been removed to keep the focus strictly on the technical contributions.
>
> #### 3. Comparison with MIP algorithms:
>
> We appreciate the request for clarification. The comparison of OR-Tools optimization convergence and our foundation model are shown in Fig. 8. While MIP can achieve the best possible results, it typically requires several hours of computation. In contrast, **our models can generate solutions within seconds that match the quality of MIP solutions requiring thousands of seconds to compute.** We have added corresponding visualized experimental results and discussion in the revised version to emphasize this trade-off. The parameters employed by OR-tools are provided in the Appendix A.2.
>
> We thank the reviewer again for their constructive feedback and for acknowledging the significance of our contributions.

---

### Meta-Review · Area_Chair_Wb89 · 2026-01-03

**Summary:**

This paper introduces how to use decoder-only architecture to solve industrial scheduling tasks. After the reviewing stages, this paper received 4 reviews, consisting of two positive and two negaive comments. The reviewer concerns can be summaried as:

- Missing DAG Context (Reviewer **1cih**)
- Overlook and Generalization on Real-World Industrial Application (Reviewer **1cih**, **F3mR**)
- LLM Adaptation (Reviewer **1cih**)
- Unclear clarification (Reviewer **F3y3**)
- $F(S)$ in the formula (Reviewer **F3y3**)
- Other RL methods

After the rebuttal stage, authors have provided response to each reviewer but still remain two ongoing cases: limitations on real-world industrial scenarios and experiments on new RL methods. After reviewing authors' response, it can be understandable that applying tree-based search algorithm may take much computations. But the concerns about application on real-world scenarios still remains. Therefore, considering the ICLR is highly competitive, this paper is not chosen for acceptance in this time.

**Reviewer Concerns:**

After the discussion stage, authors have given response for each reviewer, including DAG case in the proposed method, more explaination about unclear parts in this paper. These response could address some concerns from reviewers. After that, there still have two ongoing cases: limitations on real-world industrial scenarios and experiments on new RL methods. Author claim they have provided partial DAG cases and can handle simpler cases, but still cannot process more complex cases, which may be one problem. For RL part, authors mentioned that tree-based search may require large computations. It can be understandable but the concern about application on real-world industrial scenarios still remains.

**Reviewer Scores:**

Authors have provided responses to all reviewers. Some partial concerns from Reviewer **1cih** (e.g., DAG cases) and **F3y3** (Clear clarifications) may be addressed but still remain concerns about limitations on real-world industrial scenarios. Therefore, these reviewers possibly still keep their score.

---

### Decision · Program_Chairs · 2026-01-26

Reject